# The utilization of advance telemetry to investigate critical physiological parameters including electroencephalography in cynomolgus macaques following aerosol challenge with eastern equine encephalitis virus

John C. Trefry [1], Franco D. Rossi[1], Michael V. Accardi[2], Brandi L. Dorsey[1], Thomas R. Sprague [1], Suzanne E. Wollen-Roberts [1], Joshua D. Shamblin[1], Adrienne E. Kimmel [1], Pamela J. Glass [1], Lynn J. Miller[3], Crystal W. Burke[1], Anthony P. Cardile [4], Darci R. Smith [1], Sina Bavari[1], Simon Authier[2], William D. Pratt [1], Margaret L. Pitt [1]*, Farooq Nasar [1]*

**1** Virology Division, United States Army Medical Research Institute of Infectious Diseases, Frederick, Maryland, United States of America, **2** Charles River (formerly Citoxlab North America), Laval, Canada, **3** Veterinary Medicine Division, United States Army Medical Research Institute of Infectious Diseases, Frederick, Maryland, United States of America, **4** Division of Medicine, United States Army Medical Research Institute of Infectious Diseases, Frederick, Maryland, United States of America

* margaret.l.pitt.civ@mail.mil (MLP); fanasar@icloud.com (FN)

## Abstract

Most alphaviruses are mosquito-borne and can cause severe disease in humans and domesticated animals. In North America, eastern equine encephalitis virus (EEEV) is an important human pathogen with case fatality rates of 30–90%. Currently, there are no therapeutics or vaccines to treat and/or prevent human infection. One critical impediment in countermeasure development is the lack of insight into clinically relevant parameters in a susceptible animal model. This study examined the disease course of EEEV in a cynomolgus macaque model utilizing advanced telemetry technology to continuously and simultaneously measure temperature, respiration, activity, heart rate, blood pressure, electrocardiogram (ECG), and electroencephalography (EEG) following an aerosol challenge at 7.0 log$_{10}$ PFU. Following challenge, all parameters were rapidly and substantially altered with peak alterations from baseline ranged as follows: temperature (+3.0–4.2˚C), respiration rate (+56–128%), activity (-15-76% daytime and +5–22% nighttime), heart rate (+67–190%), systolic (+44–67%) and diastolic blood pressure (+45–80%). Cardiac abnormalities comprised of alterations in QRS and PR duration, QTc Bazett, T wave morphology, amplitude of the QRS complex, and sinoatrial arrest. An unexpected finding of the study was the first documented evidence of a critical cardiac event as an immediate cause of euthanasia in one NHP. All brain waves were rapidly (∼12–24 hpi) and profoundly altered with increases of up to 6,800% and severe diffuse slowing of all waves with decreases of ∼99%. Lastly, all NHPs exhibited disruption of the circadian rhythm, sleep, and food/fluid

**Data Availability Statement:** All relevant data are within the manuscript and its Supporting Information files.

**Funding:** This study was supported by a grant from Medical Countermeasure Systems-Joint Vaccine Acquisition Program [Grant #A5XA0A7444182001 (FN and MLP)]. The funders had no role in study design, data collection and analysis, decision to publish, or preparation of the manuscript.

**Competing interests:** The authors have declared that no competing interests exist.

intake. Accordingly, all NHPs met the euthanasia criteria by ~106–140 hpi. This is the first of its kind study utilizing state of the art telemetry to investigate multiple clinical parameters relevant to human EEEV infection in a susceptible cynomolgus macaque model. The study provides critical insights into EEEV pathogenesis and the parameters identified will improve animal model development to facilitate rapid evaluation of vaccines and therapeutics.

## Author summary

In North America, EEEV causes the most severe mosquito-borne disease in humans highlighted by fatal encephalitis and permeant debilitating neurological sequelae in survivors. The first confirmed human cases were reported more than 80 years ago and since then multiple sporadic outbreaks have occurred including one of the largest in 2019. Unfortunately, most human infections are diagnosed at the on-set of severe neurological symptoms and consequently a detailed disease course in humans is lacking. This gap in knowledge is a significant obstacle in the development of appropriate animal models to evaluate countermeasures. Here, we performed a cutting-edge study by utilizing a new telemetry technology to understand the course of EEEV infection in a susceptible macaque model by measuring multiple physiological parameters relevant to human disease. Our study demonstrates that the infection rapidly produces considerable alterations in many critical parameters including the electrical activity of the heart and the brain leading to severe disease. The study also highlights the extraordinary potential of new telemetry technology to develop the next generation of animal models to comprehensively investigate pathogenesis as well as evaluate countermeasures to treat and/or prevent EEEV disease.

## Introduction

The genus *Alphavirus* in the family *Togaviridae* is comprised of small, spherical, enveloped viruses with genomes consisting of a single stranded, positive-sense RNA ~11–12 kb in length. Alphaviruses comprise 31 recognized species classified into eleven complexes based on antigenic and/or genetic similarities [1–5]. The two aquatic alphavirus complexes [Salmon pancreatic disease virus (SPDV) and Southern elephant seal virus (SESV)] are not known to utilize arthropods in their transmission cycles, whereas all of the remaining complexes [Barmah Forest, Ndumu, Middelburg, Semliki Forest, Venezuelan (VEE), eastern (EEE), western equine encephalitis (WEE), Trocara, and Eilat], consist of arboviruses that almost exclusively utilize mosquitoes as vectors [6]. Mosquito-borne alphaviruses infect diverse vertebrate hosts including equids, birds, amphibians, reptiles, rodents, pigs, nonhuman primates (NHPs), and humans [6].

The ability to infect both mosquitoes and vertebrates enables the maintenance of alphaviruses in natural endemic transmission cycles that occasionally spillover into the human population and cause disease. Infections with Old World alphaviruses such as chikungunya, o'nyong-nyong, Sindbis, and Ross River are rarely fatal but disease is characterized by rash and debilitating arthralgia that can persist for months or years [6]. In contrast, New World alphaviruses such as eastern (EEEV), western (WEEV), and Venezuelan equine encephalitis virus (VEEV) can cause fatal encephalitis [6].

Of the New World alphaviruses, EEEV is of foremost importance in North America. EEEV is comprised of four lineages; one North American (NA) and three South American [7,8]. The

NA lineage is associated with severe human disease, and is endemic in the eastern United States and Canada, and the Gulf coast of the United States [6]. The main transmission cycle is between passerine birds and *Culiseta melanura* mosquitoes, with enzootic foci concentrated in the Mid-Atlantic, New England, Michigan, Wisconsin, and Florida [6,9]. This cycle can spill-over into humans and domesticated animals to cause severe disease with human and equid case-fatality rates of 30–90% and >90%, respectively [6,10]. Human survivors can suffer from debilitating and permanent long-term neurological sequelae at rates of 35–80% [6,10]. In addition to natural infections, many properties of EEEV including ease of isolation from nature and amplification in tissue culture, high virus titers, virus stability, high infectivity and uniform lethality via aerosol route in non-human primates are conducive to weaponization. These properties facilitated the development of EEEV as a potential biological weapon during the Cold War by the United States and the former Union of Soviet Socialist Republics (USSR) [11,12]. These traits have also led to the assignment of EEEV to the NIAID category B list and as a select agent. Currently, there are no licensed therapeutics and/or vaccines to treat or prevent EEEV infection and the U.S. population remains vulnerable to bioterror event/s or natural disease outbreaks.

In order to develop effective therapeutic and/or vaccine countermeasures, various rodent and NHP models have been utilized to recapitulate various aspects of human disease. Among these models, aerosol infection of the cynomolgus macaques can produce alterations in blood chemistry and hematology, febrile illness, viremia, neurological disease, and lethality [13–15]. However, the data in the model are limited and requires further examination to gain insights into EEEV clinical disease course and pathogenesis. In this study, we investigated disease course in the cynomolgus macaque model following an aerosol challenge with EEEV utilizing state of the art telemetry to measure clinical signs including temperature, activity, respiration, heart rate, blood pressure, electrocardiogram (ECG), and electroencephalography (EEG).

## Materials and methods

### Ethics statement

This work was supported by an approved USAMRIID Institute Animal Care and Use Committee (IACUC) animal research protocol. Research was conducted under an IACUC approved protocol in compliance with the Animal Welfare Act, PHS Policy, and other Federal statutes and regulations relating to animals and experiments involving animals. The facility where this research was conducted is accredited by the Association for Assessment and Accreditation of Laboratory Animal Care (AAALAC International) and adheres to principles stated in the Guide for the Care and Use of Laboratory Animals, National Research Council, 2011 [16].

### Virus and cells

Eastern equine encephalitis virus isolate V105-00210 was obtained from internal USAMRIID collection. The virus (Vero-1) was received from the Centers for Disease Control and Prevention (CDC) Fort Collins, CO [17]. The virus stock was passed in Vero-76 cells (American Type Culture Collection, ATCC; Bethesda, MD) twice for the production of Master (Vero-2) and Working (Vero-3) virus stocks. The virus stock was deep sequenced to verify genomic sequence and to ensure purity. In addition, the stock was tested and determined to be negative for both endotoxin and mycoplasma.

Vero-76 cells were propagated at 37˚C with 5% $CO_2$ in Dulbecco's Minimal Essential Medium (DMEM) (CellGro) containing 2% (v/v) fetal bovine serum (FBS) (Hyclone), sodium

pyruvate (1 mM) (CellGro), 1% (v/v) non-essential amino acids (CellGro), and 50 µg/mL gentamicin (Invitrogen).

## Nonhuman primate study design

Four (2 males, 2 females) cynomolgus macaques (*Macaca fascicularis*) of Chinese origin ages 5–8 years and weighing $\sim$3–9 kg were obtained (Covance). All NHPs were prescreened and determined to be negative for Herpes B virus, simian T-lymphotropic virus 1, simian immunodeficiency virus, simian retrovirus D 1/2/3, tuberculosis, *Salmonella* spp., *Campylobacter* spp., hypermucoviscous *Klebsiella* spp., and *Shigella* spp. NHPs were also screened for the presence of neutralizing antibodies to EEEV, VEEV IAB, and WEEV by plaque reduction neutralization test ($PRNT_{80}$).

## Telemetry devices and data collection

All NHPs were implanted with devices by Covance/DSI and 4-weeks post-implantation the NHPs were transported to USAMRIID. NHPs were implanted with a Data Sciences International (DSI) PhysioTel Digital M11 and two DSI PhysioTel Digital M01 implants. Each M01 device was dedicated to each hemisphere of the brain to measure EEG activity. The devices were implanted at left and right scapula and the leads were placed intracranially. The M11 implant was utilized to measure temperature, activity, respiration and heart rates, blood pressure, and ECG. Following implantation, the NHPs completely recovered after 2–4 weeks.

Implanted animals were placed in individual cages with a single DSI TRX-1 receiver per cage with additional TRX-1 receivers placed in the study room for redundancy. These receivers were connected via Cat 5e cable to communication link controllers. The digital data was routed to data acquisition computers, which captured and archived the digital data using the Notocord-hem Evolution software platform (Version 4.3, Notocord Inc., Newark, NJ).

The telemetry devices were activated in the NHPs and pre-challenge baseline values for each parameter (temperature, activity, respiration, heart rate, blood pressure, ECG, and EEG) were obtained for five days. All physiological parameters were sampled at various rates; activity and temperature (1 Hz), blood pressure and ECG (500 Hz), and EEG (1,000 Hz). Respiration and heart rates were derived utilizing NOTOCORD software (NOTOCORD Systems, Instem Company, Le Pecq, France). EEG analysis was performed using NeuroScore software version 3.1 (Data Sciences International, St. Paul, Minnesota, USA).

## Establishing baseline for each physiological parameter

To establish a baseline of each parameter for individual animals, telemetry data was collected continuously for five days prior to challenge. Data for each animal and parameter was utilized to generate a 0.5-hr interval by averaging up to 1,800 data points. Subsequently, a 48-point reference baseline for a 24-hr period was generated by averaging time-matched five previously calculated baseline values. Baseline 0.5-hr averages and standard deviations (SD) were generated. Analysis was also performed in 12-hr day/nighttime intervals. Daytime and nighttime are defined as 6 am to 6 pm and 6 pm to 6 am, respectively. To generate a 12-hr average, all raw data points with respective times were averaged to generate each day/nighttime values. All comparisons between pre- and post-challenge were time matched.

Following determination of baseline values, NHPs were challenged with a target dose of 7.0 $\log_{10}$ PFU of EEEV via the aerosol route. The NHPs were observed for signs of clinical disease and data for each parameter (temperature, activity, respiration and heart rates, blood pressure, ECG, and EEG) was obtained. Time-matched comparisons were made between pre-challenge baseline and post-challenge values.

## Aerosol challenge

NHPs were exposed to the target inhalation dose of 7.0 $\log_{10}$ PFU of EEEV in the head-only Automated Bioaerosol Exposure System (ABES-II). The virus stock at 9.3 $\log_{10}$ PFU/mL was diluted to 9.0 $\log_{10}$ PFU/mL and utilized in the nebulizer. The inhalation challenge was generated using a Collison Nebulizer to produce a highly respirable aerosol (flow rate 7.5±0.1 L/minute). The system generated a target inhalation of 1 to 3 μm mass median aerodynamic diameters determined by TSI Aerodynamic Particle Sizer. Samples of the pre-spray suspension and inhalation collected from the exposure chamber using an all-glass impinger (AGI) during the challenge were analyzed by plaque assay to determine the inhaled PFU. The inhalation challenge dose for each NHP was calculated from the minute volume determined with a whole-body plethysmograph box using Buxco XA software. The total volume of inhaled dose was determined by the exposure time required to deliver the estimated inhaled dose. Individual NHPs were challenged successively in the ABES-II.

## Post-exposure monitoring and score criteria

NHP observations began five days prior to aerosol exposure to obtain baseline data. Following aerosol challenge all NHPs were monitored daily via continuous 24-hr remote monitoring to limit room entries. Clinical signs of disease were observed and a score for each NHP was determined by evaluating three parameters; neurological score, temperature, and responsiveness score. The neurological score scale was as follows: 0 = normal; 1 = mild and infrequent tremors, 2 = hyperactivity, infrequent tremors, 3 = constant and repetitive tremors, and 10 = unresponsive. Temperature score scale was as follows; 0 = normal baseline, 1 = 1˚C above or below baseline, 2 = 2˚C above or below baseline, 3 = 3˚C above or below baseline, 10 = 4˚C above or below baseline. Responsive score scale was as follows; 0 = normal, 1 = mild unresponsiveness, 2 = moderate unresponsiveness, 3 = severe unresponsiveness, and 10 = unresponsive. NHPs with a total score ≥10 met the euthanasia criteria.

## Tissue preparation

All tissues and fluids were collected at the time of euthanasia and frozen. For quantification of virus, frozen samples of brain (frontal cortex), olfactory bulb, cervical spinal cord, and heart were thawed, weighed, and suspended in 1X PBS to generate 10% (w:v) tissue suspensions using a Mixer Mill 300 (Retsch, Haan, Germany). Tissue homogenates were centrifuged at 5,000 x *g* for 5 mins and clarified supernatants were used immediately for plaque assay as described below. Cerebral spinal fluid (CSF), plasma, and serum samples were thawed and used immediately in plaque assays.

## Plaque assay

ATCC Vero 76 cells were seeded overnight on 6-well tissue culture plates to achieve 90–95% confluence. Triplicate wells were infected with 0.1-ml aliquots from serial 10-fold dilutions in Hanks' Balanced Saline Solution (HBSS) and virus was adsorbed for 1 hr at 37˚C, 5% $CO_2$. After incubation, cells were overlaid with Eagle's Basal Medium (BME) (Gibco A15950DK) containing 0.6% agarose supplemented with 10% heat-inactivated FBS, 2% Penicillin/Streptomycin (10,000 IU/mL and 10,000 μg/mL, respectively), and incubated for 24 hr at 37˚C, 5% $CO_2$. A second agarose overlay, prepared as described above, containing 5% neutral red vital stain (Gibco 02-0066DG) was added to the wells and incubated for 18–24 hr for visualization of plaques. Plaques were counted and expressed in either plaque forming units (PFU) per mL (PFU/mL) or PFU/g of tissue.

### Plaque reduction neutralization test (PRNT$_{80}$)

Serum samples were heat-inactivated at 56°C for 30 mins. Samples were serially diluted 2-fold starting at 1:10 and were mixed with equal volumes of medium containing ~2,000 PFU/mL of virus and incubated at 37°C, 5% $CO_2$. Following incubation, six-well plates containing monolayers of Vero-76 cells were infected with 100μL of virus-serum mixtures in triplicates and incubated at 37°C, 5% $CO_2$ for ~1 hr. Following incubation, a secondary agarose overlay containing 5% neutral red vital stain was added to the wells and incubated for ~24 hr for visualization of plaques. Plaques were counted, PRNT titers were calculated and expressed as the reciprocal of serum dilution yielding a >80% reduction (PRNT$_{80}$) in the number of plaques. The limit of detection in PRNT$_{80}$ assay is <1:20. All samples were analyzed three times in the assay.

### Statistics

All comparisons between pre- and post-challenge were time matched. GraphPad Prism version 7.00 for Windows (GraphPad Software, La Jolla, California, USA) software was utilized for statistical analysis. Significant differences in each parameter were determined using one-way ANOVA followed by a Tukey Test.

## Results

### EEEV challenge study design, survival, and detection of infectious virus in tissues at terminal time point

Four macaques (2 males and 2 females) were implanted with telemetry devices to continuously and simultaneously monitor physiological parameters for the duration of the study. Baseline for each parameter in each NHP was determined by obtaining data for five daytime and night-time cycles. Following establishment of baseline, the NHPs were challenged via the aerosol route with EEEV V105 strain at a target dose of 7.0 log$_{10}$ PFU (Fig 1). The NHPs received a virus dose ranging between 6.4–6.8 log$_{10}$ PFU (Fig 2A). Following challenge, the NHPs were observed for signs of clinical disease and each NHP was assigned a score comprising of alterations in temperature, responsiveness, and neurological signs. NHPs with a score of ten or higher met the euthanasia criteria. All NHPs exhibited signs of clinical disease by ~48–72 hours post-infection (hpi) (S1 Fig). NHP #1 and #2 exhibited rapid increase in scores and met the euthanasia criteria by ~106–120 hpi (Figs 2B and S1). The two remaining NHPs displayed a similar initial rise in scores, followed by a transient decline and a rapid progression to severe disease at ~140 hpi (Figs 2B and S1).

Various NHP tissues were collected at the time of euthanasia and EEEV was quantitated via plaque assay. Virus was titrated from samples including serum, plasma, brain (frontal cortex), olfactory bulb, cervical spinal cord, and heart of each NHP (Fig 3). Cerebrospinal fluid (CSF) was collected and titrated for NHPs #1 and #2 (Fig 3). Infectious virus was detected in serum and plasma of only NHP #1 with titers of ~3.5 and ~3.2 log$_{10}$ PFU, respectively (Fig 3). In contrast, EEEV was detected in brain and olfactory bulb in all four NHPs with titers ranging from ~4.1 to ~7.9 log$_{10}$ PFU (Fig 3). The cervical spinal cord of NHP #2 and #4 had virus titers of ~7.6 and ~4.1 log$_{10}$ PFU, respectively (Fig 3). Virus was detected in the heart tissue of only one NHP (NHP #3) with a titer of ~4.0 log$_{10}$ PFU (Fig 3). Lastly, the virus was present in CSF of NHP #1 and #2 with titers of ~5.9 and ~5.5 log$_{10}$ PFU, respectively (Fig 3).

### Alteration of animal behavior

The NHPs were observed were continuously monitoring with limited disruptions due to human activity. This provided a rare opportunity to study the impact of EEEV infection on

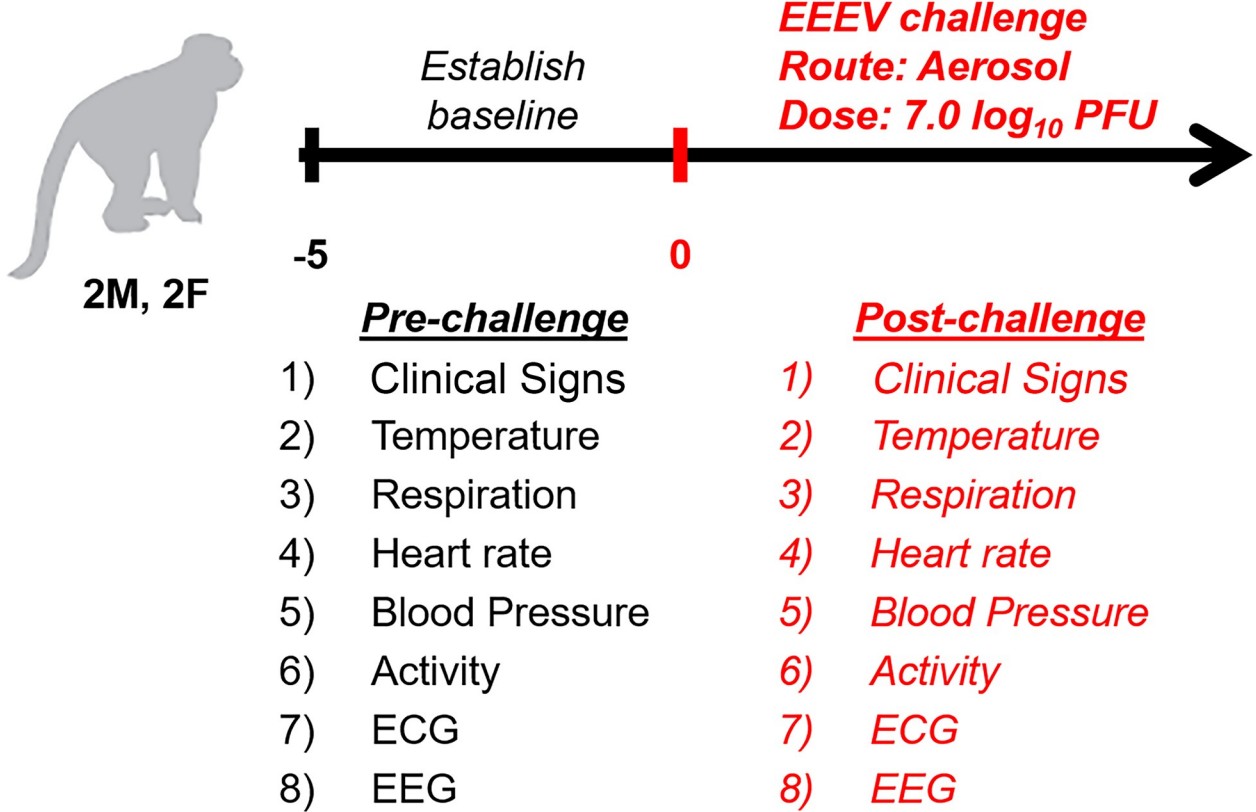

**Fig 1. Experimental design of the NHP study.** All parameters were continuously and simultaneously measured throughout the study. Baseline was established for each parameter for an individual NHP by measuring 5 daytime (6 am-6 pm) and nighttime (6 pm– 6 am). All comparisons of post-challenge data was time-matched to pre-challenge baseline.

animal behavior and determine the onset of neurological signs. The baseline behavior was determined for five days prior to challenge for each individual NHP for day and night times. Daytime and nighttime were defined as 6 am to 6pm and 6pm to 6 am, respectively. Alteration of animal behavior were evaluated by observing three parameters; sleep, activity, and food/fluid consumption. Alteration of the circadian rhythm could be detected as early as 24 hpi in NHP #4 and ~42–54 hpi in the remaining three NHPs (S1 Table). The disruption of circadian rhythm was characterized by a decrease in sleep at night with a concomitant increase in nighttime activity (S1 Table). The following daytime period was characterized by a decrease in daytime activity with short periods of sleep (S1 Table). However, following the onset of clinical signs, all NHPs exhibited a rapid progression to no or minimal sleep for the remainder of the study (S1 Table). Similar to sleep, food/fluid consumption also decreased between ~43–90 hpi in three of the four NHPs with minimal food/fluid consumption ~15–36 hrs prior to euthanasia. Lastly, the onset of overt seizures was observed within last ~7 hrs of the study in three of the four NHPs (S1 Table).

## Neutralizing antibody response

The presence of neutralizing antibodies was measured via $PRNT_{80}$ at days -7, 0, and terminal time points (S2 Table). None of the NHPs had detectable neutralizing antibody titers prior to or at the time of challenge and were assigned the limit of detection of the assay ($<$1:20) (S2 Table). At the time of euthanasia, NHPs #1 and #2 did not have any detectable neutralizing

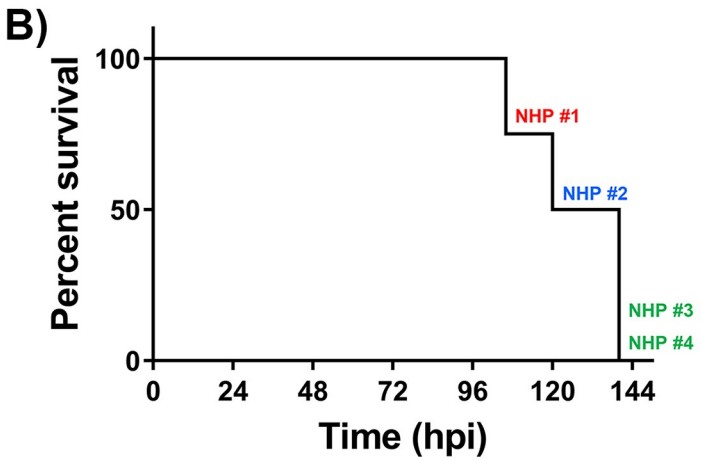

**A)**

| Animal ID | Sex | Age (yrs) | Weight (kg) | Target Dose (log$_{10}$ PFU/NHP) | Dose Administered (log$_{10}$ PFU/NHP) |
|-----------|-----|-----------|-------------|----------------------------------|----------------------------------------|
| NHP #1 | M | 8 | 7 | 7.0 | 6.6 |
| NHP #2 | F | 5 | 3 | 7.0 | 6.6 |
| NHP #3 | F | 6 | 4 | 7.0 | 6.4 |
| NHP #4 | M | 5 | 9 | 7.0 | 6.8 |

**B)**

**Fig 2.** Administered aerosol EEEV dose (A) and survival of the NHPs (B).

antibody response (S2 Table). In contrast, both NHP #3 and #4 had PRNT$_{80}$ titers of 1:40 and 1:80, respectively (S2 Table).

## Temperature

The baseline temperature was determined for each NHP for five days and data was analyzed in 0.5- (Fig 4A) and 12-hr (Fig 4B) intervals. All post-challenge comparisons were time-matched to pre-challenge baseline values. The average temperature prior to challenge ranged from ~36.3–38°C (Fig 4A). Following challenge, an increase in temperature was observed within ~26–56 hpi in all four NHPs (Fig 4A, S3 Table). The onset of fever (>1.5°C above baseline) was at ~53–61 hpi and it remained considerably elevated for duration of ~47–53 hrs (Fig 4A, S3 Table). The peak temperature ranged from 40.1–41°C, and peak magnitude of fever ranged from ~3.0–4.2°C and ~2.2–3.8°C for 0.5- and 12-hr interval analyses, respectively (Fig 4A and 4B, S3 Table). In both analyses, NHPs displayed the highest elevation in temperature at night time (Fig 4A and 4B). Three of the NHPs exhibited hyperpyrexia (>3.0°C above baseline) for a duration of ~11–22 hrs (Fig 4A and 4B). Following the sustained fever, a decline in temperature was observed in the last ~10–24 hrs prior to euthanasia in all four NHPs (Fig 4A). NHPs #3 and 4 exhibited a rapid decline in temperature ~4–6 hrs prior to euthanasia with NHP #3 displaying a decline of 3.3°C (Fig 4A).

## Respiration rate

Similar to temperature, respiration rate was analyzed in 0.5- (Fig 5A) and 12-hr (Fig 5B) intervals. The baseline respiration rate for each NHP ranged from ~15–32 breaths per minute (bpm) (Fig 5A). Following challenge, an increase in respiration rate was observed starting at

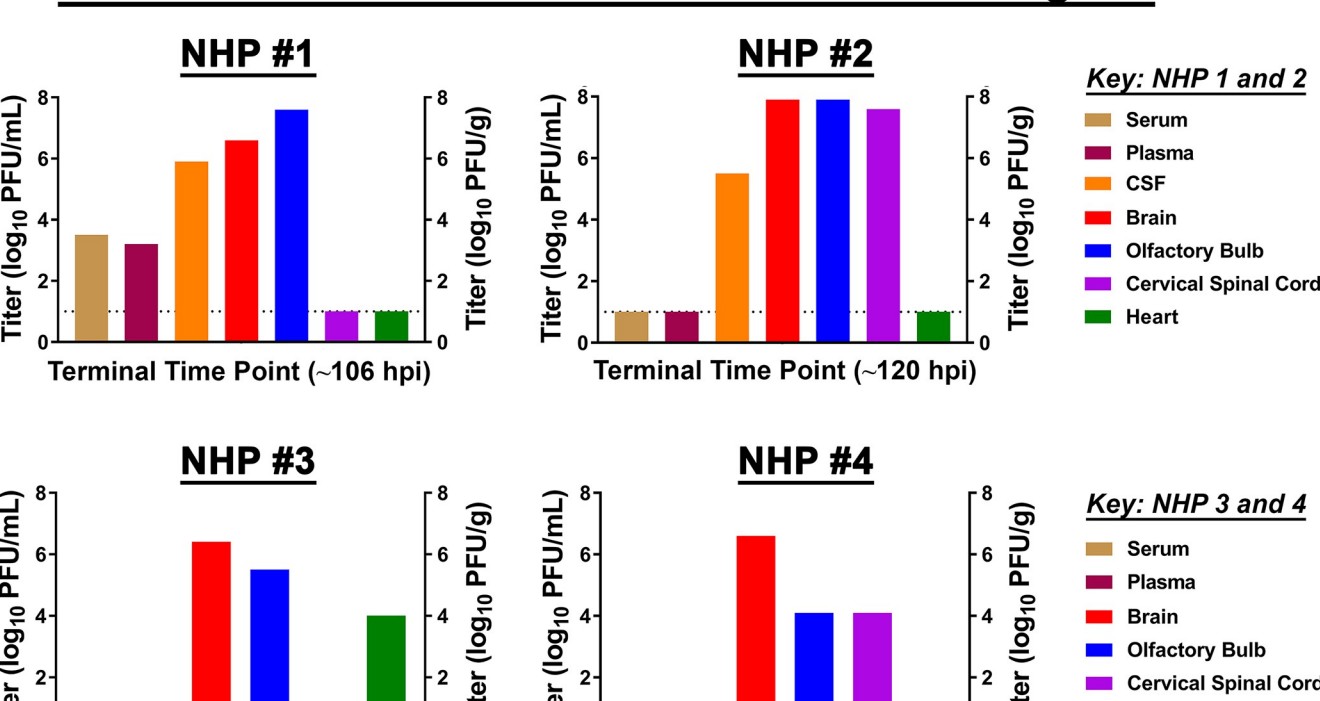

**Fig 3. Quantitation of infectious EEEV in NHP tissues collected at the time of euthanasia.** The time of tissue collection for each NHP is provided in the x-axis. Limit of detection of plaque assay is 1.0 $\log_{10}$ PFU/mL or 1.0 $\log_{10}$ PFU/g and are indicated by a dashed line.

24 hpi in all NHPs (Fig 5A and 5B). However, in contrast to temperature, the respiration rate displayed intermittent increase/decrease throughout first 96 hpi (Fig 5A). Sustained increase was observed at 100–112 hpi in all four NHPs, however the duration and magnitude of the increase differed considerably (Fig 5A). In NHPs #1 and #2, the sustained increase was observed ~3–8 hrs prior to euthanasia with peak respiration rate of 31–38 bpm, an increase of ~56–71% or ~17–22% in 0.5- and 12-hr interval analysis, respectively (Fig 5A and 5B). In contrast, NHPs #3 and #4 experienced considerably higher peak respiration rate of ~49–50 bpm for a duration of ~32–40 hrs (Fig 5A and 5B). At peak, the percent increase in respiration rate in 0.5- and 12-hr interval analysis ranged from ~105–128% and ~85–95%, respectively (Fig 5A and 5B).

## Activity

The activity of each NHP was analyzed in 6- (Fig 6A) and 12-hr (Fig 6B) intervals for both daytime and nighttime periods. Daytime activity differed markedly in the four NHPs. The average daytime activity for NHP #1 and #3 ranged from ~803–1704 units/6hrs, whereas the range for remaining two NHPs was ~410–505 units/6hrs (Fig 6A). Alteration in daytime activity could be observed within ~36–102 hpi with considerable decline in all NHPs at ~12–36 hrs prior to euthanasia (Fig 6A). The highest magnitude of decline was in NHP #1 and #3 with values of ~754–1303 units/6hrs, a decline of ~69–76% (Fig 6A). A 12-hr daytime interval analysis showed a sustained decline in activity ranging from ~64–73% in both NHPs (Fig 6B). NHP #2

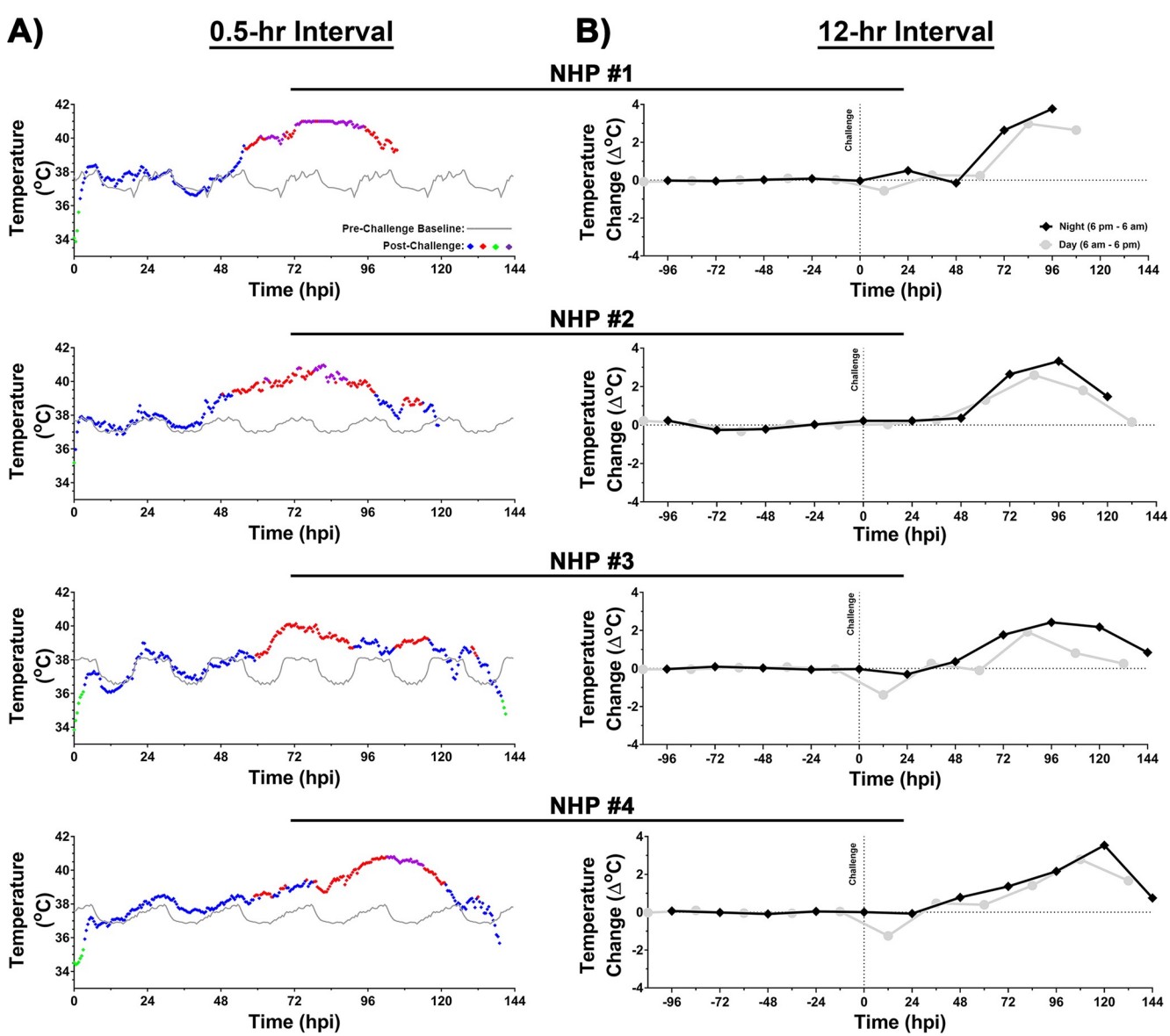

**Fig 4. Body temperature of NHPs pre- and post-EEEV challenge.** Data analysis is shown in 0.5- (A) and 12-hr (B) daytime/nighttime intervals. All NHPs were continuously monitored pre- and post-challenge. Pre-challenge baseline temperature was measured for five day/night cycles and a 0.5-hr interval baseline average was calculated by averaging raw data of five time-matched day or night time intervals. Forty-eight 0.5-hr interval averages were used to construct a baseline temperature for a day/night cycle and is shown as a grey line (A). Post-challenge values within ≤3 standard deviations (SD) are indicated with (◆), >3 SD above baseline are indicated with (◆), and >3 SD below baseline are indicated with (◆). Hyperpyrexia is indicated by (◆) (A). Temperature change in a 12-hr interval is shown in grey (daytime) and in black (nighttime) (B). All raw data from either 12-hr daytime or nighttime intervals were averaged and normalized. The last data point for NHP #3 and #4 is an average of 8 hrs (B).

and #4 exhibited a decline in daytime activity ranging ~62–118 units/6hrs, a decline of ~15–23% and a sustained decline of ~11–20% in the 12-hr interval analysis, respectively (Fig 6B).

The baseline nighttime activity of all four NHPs was comparable, ranging from 333–383 units/6hrs (Fig 6A). Similar to daytime activity, a substantial increase in nighttime activity was observed as early as 24 hpi in NHP #4 and at 6–48 hrs prior to euthanasia in the other three NHPs (Fig 6A). Both 6- and 12-hr analysis showed an increase of ~5–22% and ~3–14%, respectively (Fig 6A and 6B). The considerable increase in nighttime activity was further corroborated via continuous monitoring (S1 Table).

## Heart rate

The baseline heart rate in NHPs ranged from ~60–150 bpm (Fig 7A). Following challenge, intermittent alterations were observed within ~24 hpi in all four NHPs (Fig 7A). Elevation in nighttime heart rate was observed within ~56 hpi followed by a return to normal daytime baseline values (Fig 7A). Sustained elevated heart rate was observed ~79–105 hpi that peaked at ~24–42 hrs prior to euthanasia with values of ~185–243 bpm (Fig 7A). The peak elevations from baseline ranged from ~67–190% and ~60–134% in 0.5- and 12-hr interval analysis, respectively (Fig 7A and 7B).

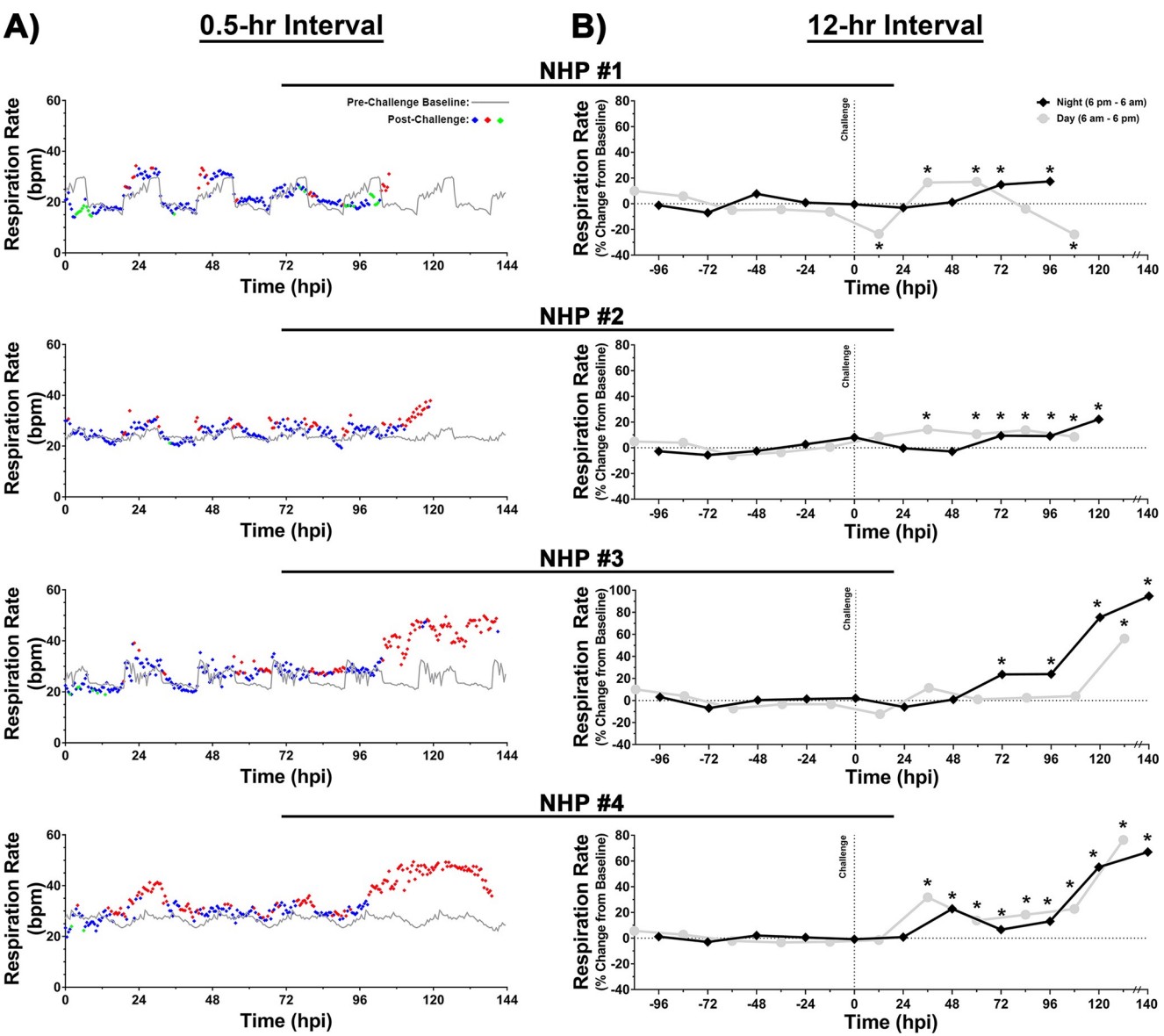

**Fig 5. Respiration rate of NHPs pre- and post-EEEV challenge.** Data analysis is shown in 0.5- (A) and 12-hr (B) daytime/nighttime intervals. All NHPs were continuously monitored pre- and post-challenge. Pre-challenge baseline respiration rate was measured for five day/night cycles and a 0.5-hr interval baseline average was calculated by averaging raw data of five time-matched day or night time intervals. Forty-eight 0.5-hr interval averages were used to construct a baseline respiration rate for a day/night cycle and is shown as a grey line (A).Post-challenge values within ≤3 standard deviations (SD) are indicated with (◆), >3 SD above baseline are indicated with (◆), and >3 SD below baseline are indicated with (◆). Percent change from baseline in 12-hr interval is shown in grey (daytime) and in black (nighttime) (B). All raw data from either 12-hr daytime or nighttime intervals were averaged and normalized. The last data point for NHP #3 and 4 is an average of 8 hrs (B). *p*-values ≤0.03 are indicated with *. bpm = breaths per minute.

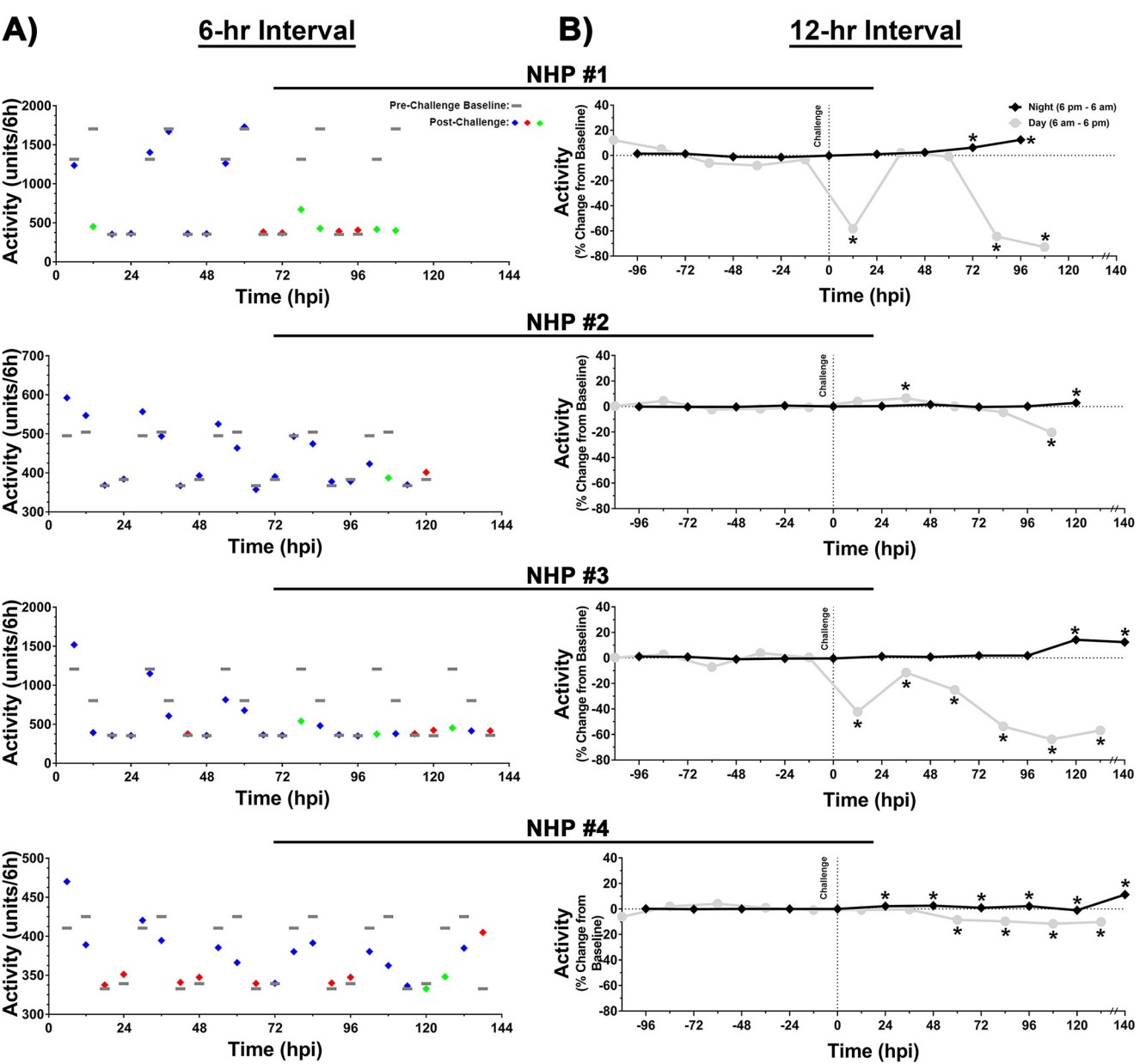

**Fig 6. Measurement of activity in NHPs pre- and post-EEEV challenge.** Data analysis is shown in 6- (A) and 12-hr (B) daytime/nighttime intervals. All NHPs were continuously monitored pre- and post-challenge. Pre-challenge baseline activity was measured for five day/night cycles and a 6-hr interval baseline average was calculated by averaging raw data of five time-matched day or night time intervals. Four 6-hr interval averages were used to construct a baseline activity for a day/night cycle and is shown as a grey line (A). Post-challenge values within ≤3 standard deviations (SD) are indicated with (♦), >3 SD above baseline are indicated with (♦), and >3 SD below baseline are indicated with (♦). Percent change from baseline in 12-hr interval is shown in grey (daytime) and in black (nighttime) (B). All raw data from either 12-hr daytime or nighttime intervals were averaged and normalized. The last data point for NHP #3 and 4 is an average of 8 hrs (B). *p*-values ≤0.041 are indicated with *.

## Blood pressure

The baseline systolic and diastolic blood pressure values ranged from ~90–116 and ~58–87 mm Hg, respectively (Figs 8A and 9A). Following challenge, both systolic and diastolic pressure intermittently increased within 24 hpi followed by a sustained elevation at ~54–76 hpi for

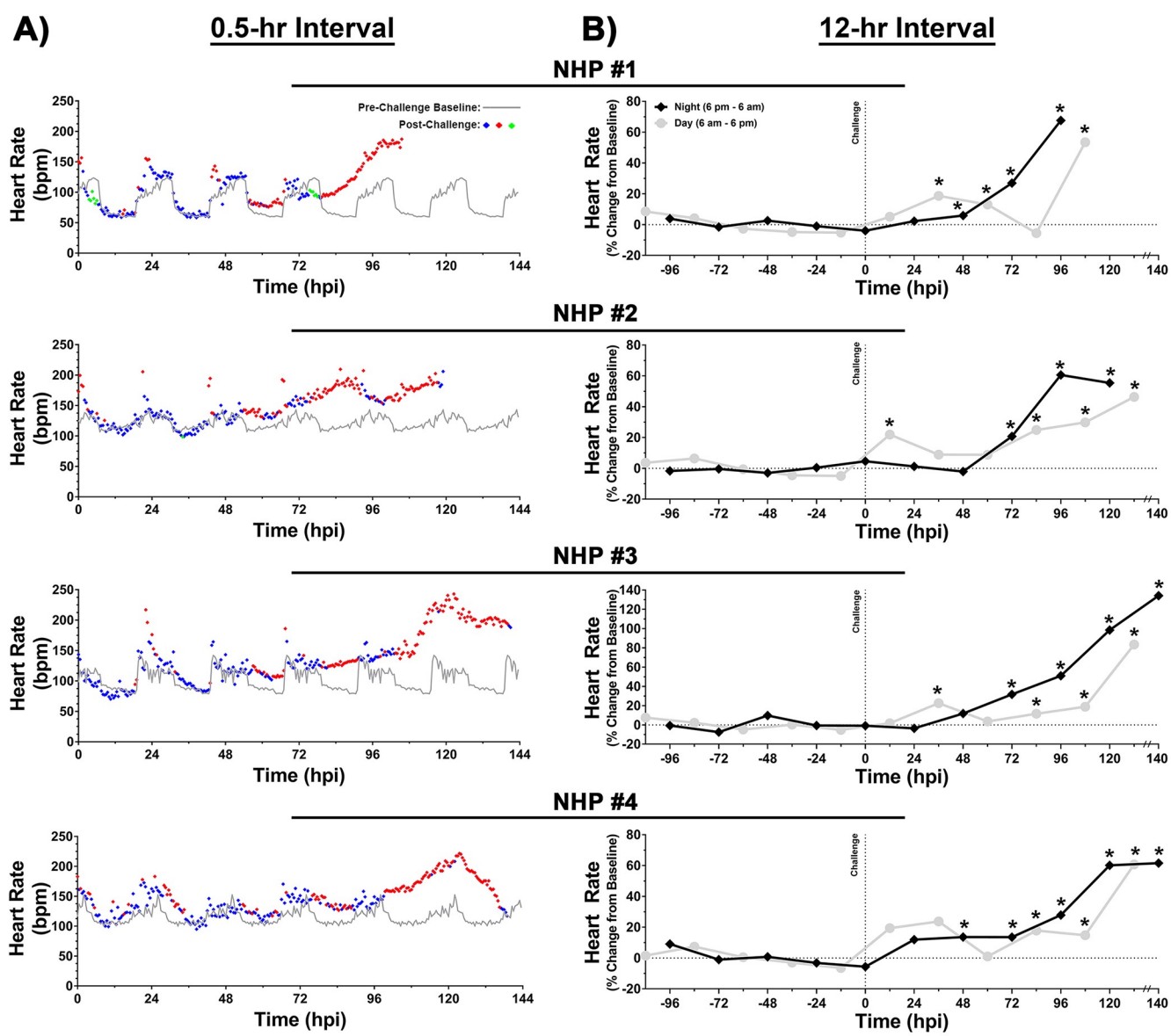

**Fig 7. Heart rate of NHPs pre- and post-EEEV challenge.** Data analysis is shown in 0.5- (A) and 12-hr (B) daytime/nighttime intervals. All NHPs were continuously monitored pre- and post-challenge. Pre-challenge baseline heart rate was measured for five day/night cycles and a 0.5-hr interval baseline average was calculated by averaging raw data of five time-matched day or night time intervals. Forty-eight 0.5-hr interval averages were used to construct a baseline heart rate for a day/night cycle and is shown as a grey line (A).Post-challenge values within ≤3 standard deviations (SD) are indicated with (♦), >3 SD above baseline are indicated with (♦), and >3 SD below baseline are indicated with (♦). Percent change from baseline in 12-hr interval is shown in grey (daytime) and in black (nighttime) (B). All raw data from either 12-hr daytime or nighttime intervals were averaged and normalized. The last data point for NHP #3 and 4 is an average of 8 hrs (B). *p*-values ≤0.035 are indicated with *. bpm = beats per minute.

a duration of ~48–80 hrs (Figs 8A and 9A). At peak, both systolic and diastolic pressures ranged from ~153–180 and ~102–128 mm Hg, respectively (Figs 8A and 9A). The peak systolic pressure values in 0.5-hr and 12-hr intervals represent percent increases of ~44–67% and ~35–57%, respectively (Fig 8A and 8B). Similarly, peak diastolic pressure values in 0.5-hr and 12-hr analysis ranged from ~45–80% and ~38–65%, respectively (Fig 9A and 9B).

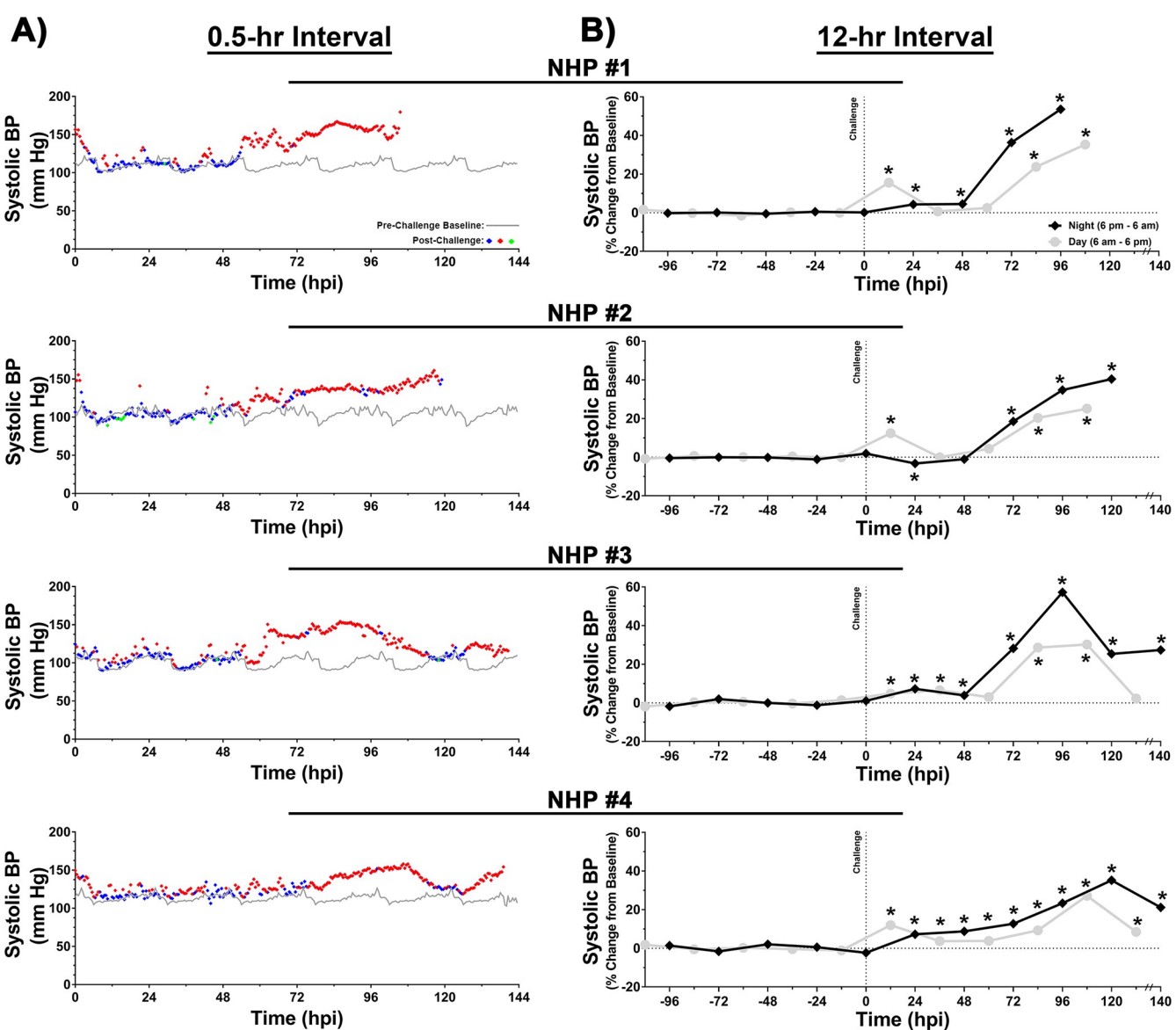

**Fig 8. Systolic blood pressure of NHPs pre- and post-EEEV challenge.** Data analysis is shown in 0.5- (A) and 12-hr (B) daytime/nighttime intervals. All NHPs were continuously monitored pre- and post-challenge. Pre-challenge baseline systolic blood pressure was measured for five day/night cycles and a 0.5-hr interval baseline average was calculated by averaging raw data of five time-matched day or night time intervals. Forty-eight 0.5-hr interval averages were used to construct a baseline systolic blood pressure for a day/night cycle and is shown as a grey line (A). Post-challenge values within ≤3 standard deviations (SD) are indicated with (◆), >3 SD above baseline are indicated with (◆), and >3 SD below baseline are indicated with (◆). Percent change from baseline in 12-hr interval is shown in grey (daytime) and in black (nighttime) (B). All raw data from either 12-hr daytime or nighttime intervals were averaged and normalized. The last data point for NHP #3 and 4 is an average of 8 hrs (B). *p*-values ≤0.028 are indicated with *.

## ECG

Intermittent changes in ECG were observed in all four NHPs following challenge as early as 6–12 hpi (S2 and S3 Figs). As expected, the sustained elevation in heart rate at ~48–72 hpi led to reduction in QTc Bazett, RR, PR, and QRS duration (S2 and S3 Figs). However, 24 hrs prior to euthanasia multiple abnormalities were observed. An increase in QRS duration was observed for ~19 hrs with peak increase of ~11 msec in NHP #1 (S2 Fig). An increase in QTc Bazett was observed in three of the four NHPs (#1, 2, and 4) for duration of ~4–12 hrs with

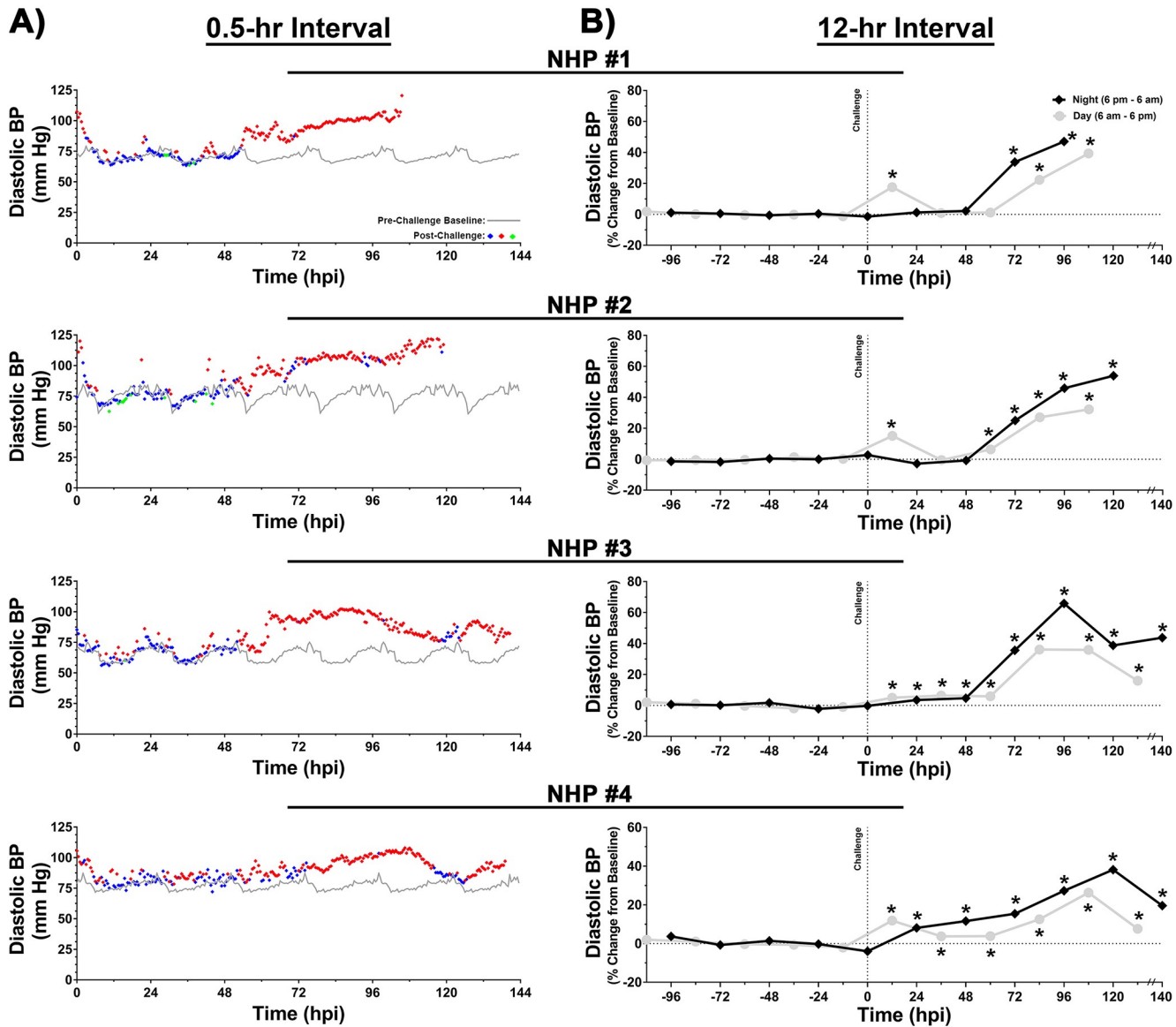

**Fig 9. Diastolic blood pressure of NHPs pre- and post-EEEV challenge.** Data analysis is shown in 0.5- (A) and 12-hr (B) daytime/nighttime intervals. All NHPs were continuously monitored pre- and post-challenge. Pre-challenge baseline diastolic blood pressure was measured for five day/night cycles and a 0.5-hr interval baseline average was calculated by averaging raw data of five time-matched day or night time intervals. Forty-eight 0.5-hr interval averages were used to construct a baseline diastolic blood pressure for a day/night cycle and is shown as a grey line (A). Post-challenge values within ≤3 standard deviations (SD) are indicated with (◆), >3 SD above baseline are indicated with (◆), and >3 SD below baseline are indicated with (◆). Percent change from baseline in 12-hr interval is shown in grey (daytime) and in black (nighttime) (B). All raw data from either 12-hr daytime or nighttime intervals were averaged and normalized. The last data point for NHP #3 and 4 is an average of 8 hrs (B). *p*-values ≤0.023 are indicated with *.

peak increase ranging from ~22–44 msec (S2 Fig). The ECG of NHPs #1, 3, and 4 displayed either intermittent or sustained elongation of PR duration (S3 Fig). NHP #4 displayed an elongation in PR duration for ~6 hrs with a peak increase of ~24 msec prior to euthanasia (S3 Fig). Alteration of T wave morphology and a decline (~1.5 mV) in the magnitude of QRS complex were both observed in NHP #2 (Fig 10). Lastly, both NHPs #1 and 4 displayed sinoatrial arrest in the last 24-hrs prior to euthanasia (Fig 10).

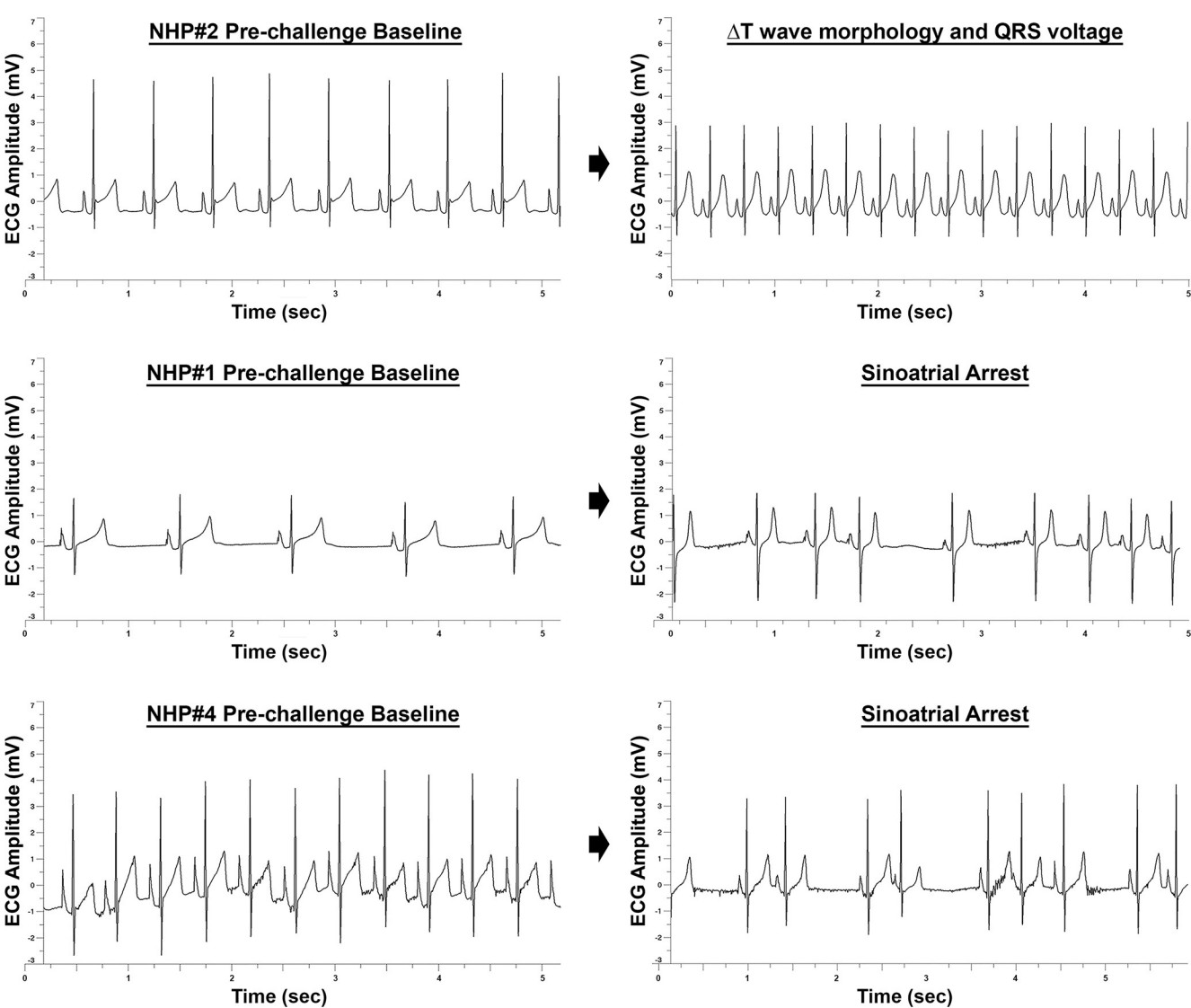

**Fig 10. ECG abnormalities in EEEV infected NHPs 24 hrs prior to euthanasia.** Representative ~5 second intervals of pre- and post-EEEV challenge are shown.

## Quantitative EEG

The EEG data for each NHP pre- and post-challenge are shown as heat maps (Figs 11 and 12). Six night and five day 12-hr intervals were used to generate a pre-challenge heat map for each NHP. In all four NHPs, the baseline heat map displayed similar patterns with the majority of values between -50 to +50%. An increase in gamma waves were observed during the daytime (generally around anticipated feeding periods and increased human activity due to arrival of NHPs in biocontainment and one day prior to challenge) in all four NHPs during the pre-challenge period attributed to electromyographic artifacts. However, post- challenge distinct patterns could be observed rapidly in all NHPs.

NHP #1 displayed an increase in the individual frequencies comprising of the beta and gamma power bands at nighttime within ~15 hr post-challenge with maximal increases of ~1,300% and ~5,200%, respectively (Fig 11). The next ~6–10 hrs were characterized by

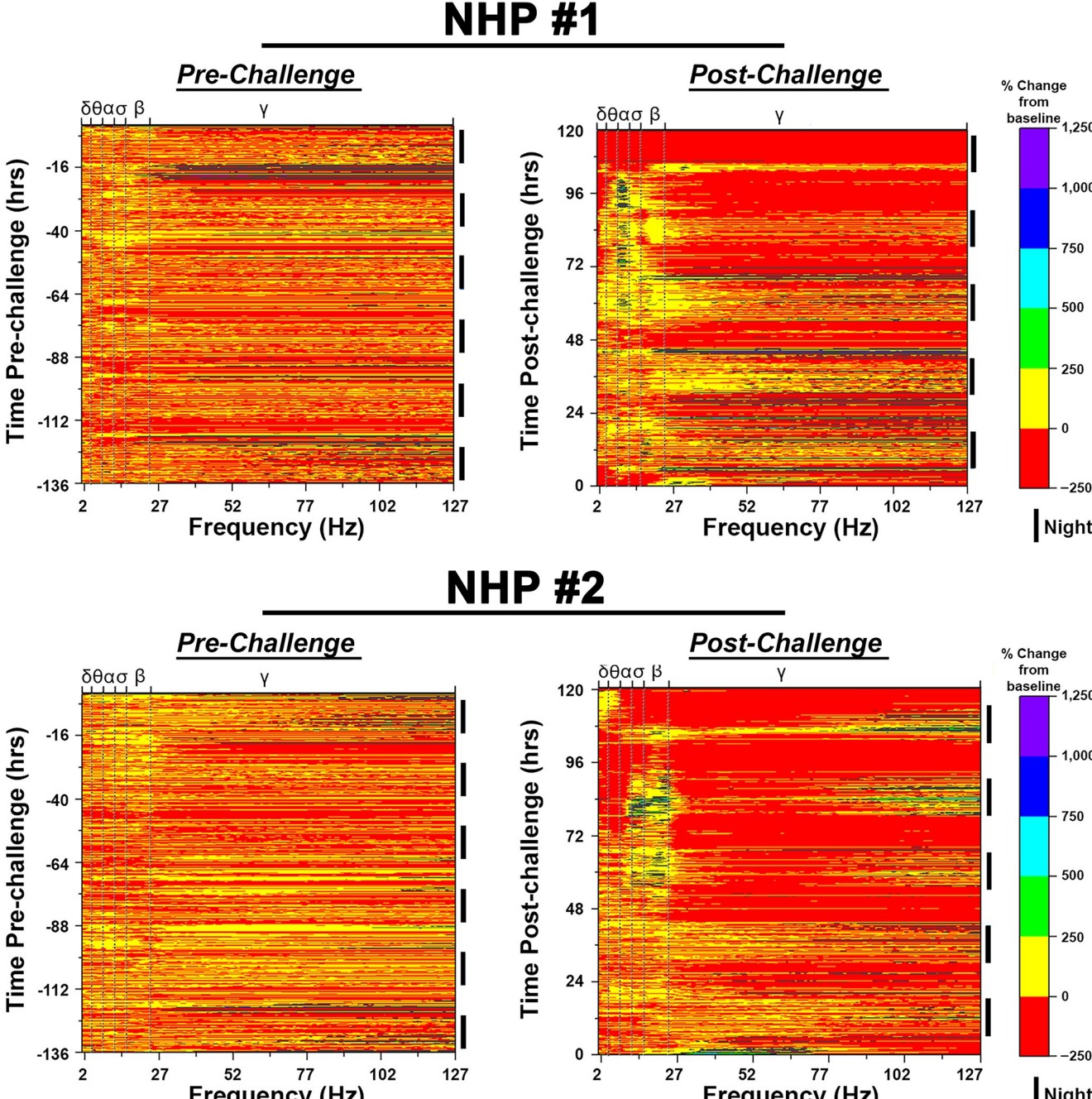

**Fig 11. Pre- and post-EEEV challenge quantitative electroencephalography (qEEG) heat maps of NHPs #1 and #2.** The top and bottom x-axes display brain waves [(delta (δ), theta (θ), alpha (α), sigma (σ), and gamma (γ)] and frequency in hertz (Hz), respectively. The left and right y-axes display time (hr) and 12-hr day/night time intervals, respectively. 12-hr nighttime is indicated by (▎) and daytime by a gap. Pre-EEEV challenge baseline (left) and post-EEEV challenge (right) heat maps are shown.

increases of up to ~800, 400, and 600% in the theta, alpha, and sigma bands, respectively, during the daytime while the NHP was awake. These increases were followed by a decline in all frequencies to near or below basal levels for ~6 hrs during the daytime (Fig 11). This pattern continued into the second night with a general increase of up to ~2,700% across all individual

frequencies within the gamma band and ~1,200% across all other frequencies (Fig 11). At 48 hpi, a prolonged decline of up to 93% was observed across all frequencies during the daytime (Fig 11). The third night was characterized by intermittent increases in high frequencies within the gamma band, however, a sustained increase of up to ~250% in all other brain waves was observed including low gamma frequencies (Fig 11). An increase of up to 6,800% in gamma frequencies was observed in the first ~3 hrs of the next daytime, followed by a sustained decline to below time-matched basal values (Fig 11). The decrease in gamma frequencies was accompanied by a simultaneous increase in theta, alpha, sigma, and low beta frequencies (in an awake NHP) with the alpha frequencies displaying the most pronounced increase with values exceeding 1,000% (Fig 11). The fourth night displayed similar patterns as the previous night with sustained increase in alpha frequencies of ~600% (Fig 11). The last daytime for this NHP was characterized by a considerable and sustained decline in delta and gamma bands to at or below basal values, with a simultaneous increase in theta, alpha, sigma, and beta frequencies (Fig 11). The magnitude of the latter waves ranged from ~80–1,400%, with alpha band displaying the largest increases (Fig 11).

NHP #2 displayed a similar pattern as NHP #1 with an early increase in the individual beta and gamma frequencies within ~12 hpi ranging from ~300 to 3,000% (Fig 11). This was followed by an elevation in delta, theta, alpha, and sigma frequencies during the daytime with increases of up to ~100 to 700% relative to time-matched baseline values (Fig 11). Concomitant to the rise in low frequencies, a decline to at or near baseline values was observed in both beta and gamma bands (Fig 11). This pattern continued into 24-hr period, however, the daytime displayed a sustained decline of both beta and gamma bands followed by an increase of up to ~750% in beta waves (Fig 11). The third night was characterized by increases in alpha, sigma, beta, and high gamma frequencies with increases of up to 2,100% in an awake NHP (Fig 11). The next daytime exhibited a prolonged decline in delta, theta, and gamma bands with a maximum reduction of nearly 99% from individual basal values (Fig 11). In contrast, sigma and alpha frequencies displayed increases of up to 660% (Fig 11). The fourth nighttime period exhibited a similar pattern to the previous nighttime. The next daytime was characterized by prolonged and considerable decline of all individual spectral frequencies to below baseline values (Fig 11). The last nighttime exhibited similar pattern to the two previous night times. Five hours prior to euthanasia was characterized by decline in all brain waves except delta and theta waves which had slight increases from ~42 to 370% in an awake NHP.

NHP #3 predominately displayed an increase in gamma frequencies in the first three nights and two daytimes with values exceeding 1,000% as compared to the time-matched baseline values (Fig 12). However, the third night also exhibited a rise in alpha, sigma, and beta frequencies not exceeding a 250% increase (Fig 12). The following daytime was predominately characterized by a prolonged and considerable decline in all individual frequencies to below time-matched baseline values. The subsequent evening and night were largely characterized by increases in alpha and gamma frequencies followed by an elevation in all frequencies except delta waves (Fig 12). The next daytime displayed a substantial reduction in all frequencies followed by a fifth nighttime with considerable increases in beta and gamma frequencies (Fig 12). The last day for NHP #3 was characterized by rise in theta, alpha, sigma, and beta waves in an awake NHP followed by a decline in all waves (Fig 12). The last nighttime showed a prolonged rise in all waves particularly in delta, theta, alpha, and sigma, in an awake NHP (Fig 12).

NHP #4 displayed an increase in both beta and gamma frequencies 12 hpi that continued for most of the nighttime and daytime following challenge (Fig 12). Surprisingly, elevation in delta, theta, alpha, and sigma were also observed within 12 hpi and were sustained until

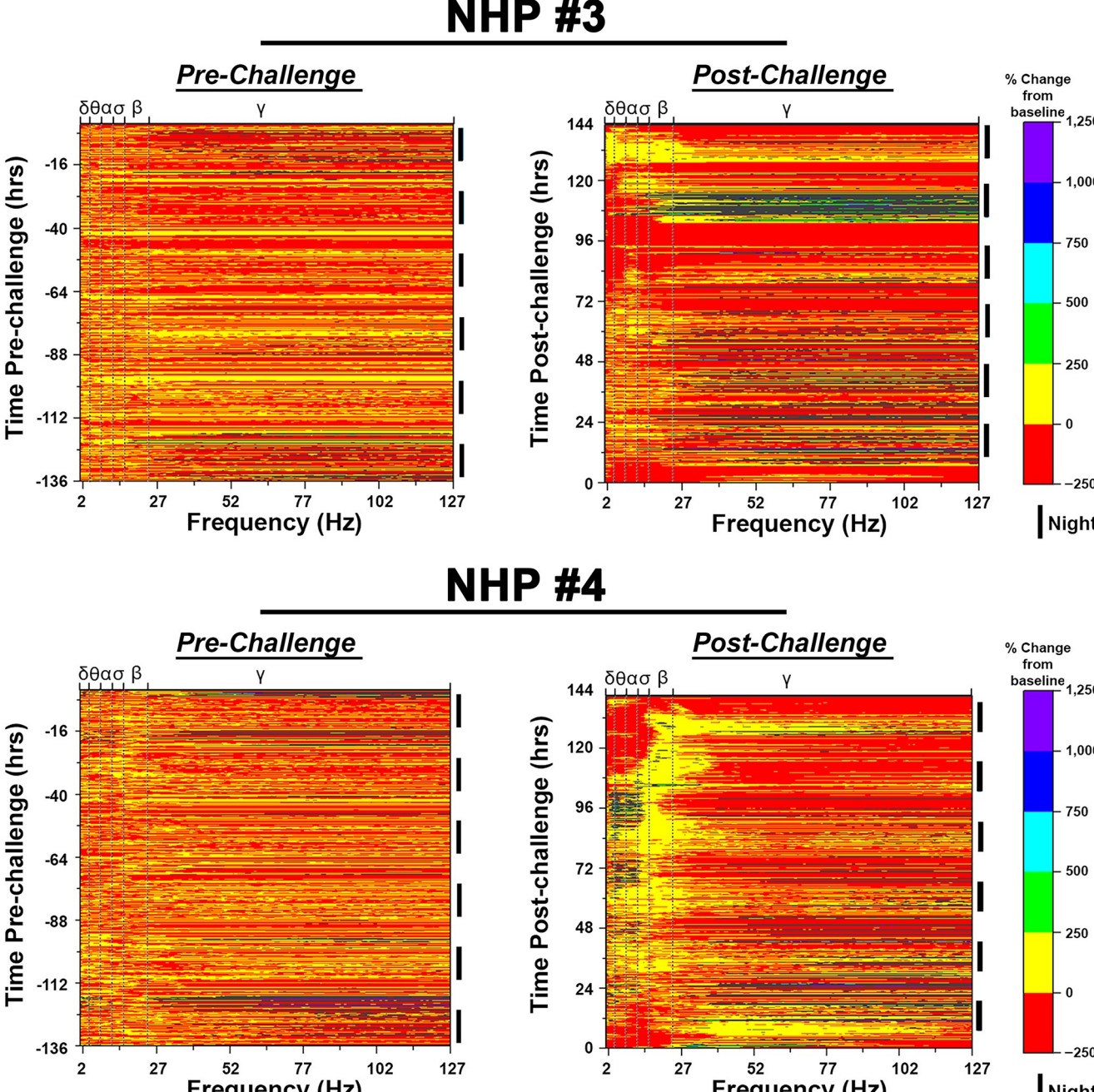

**Fig 12. Pre- and post-EEEV challenge quantitative electroencephalography (qEEG) heat maps of NHPs #3 and #4.** The top and bottom x-axes display brain waves [(delta (δ), theta (θ), alpha (α), sigma (σ), and gamma (γ)] and frequency in hertz (Hz), respectively. The left and right y-axes display time (hr) and 12-hr day/night time intervals, respectively. 12-hr nighttime is indicated by (❚) and daytime by a gap. Pre-EEEV challenge baseline (left) and post-EEEV challenge (right) heat maps are shown.

108–120 hpi with peak increases up to 1,800% in an awake NHP (Fig 12). This sustained increase over baseline was followed by a precipitous decline to below time-matched basal levels in frequencies from all four power bands from 108–136 hpi (Fig 12). Several hours prior to euthanasia, an increase in both delta and theta frequencies was observed (Fig 12).

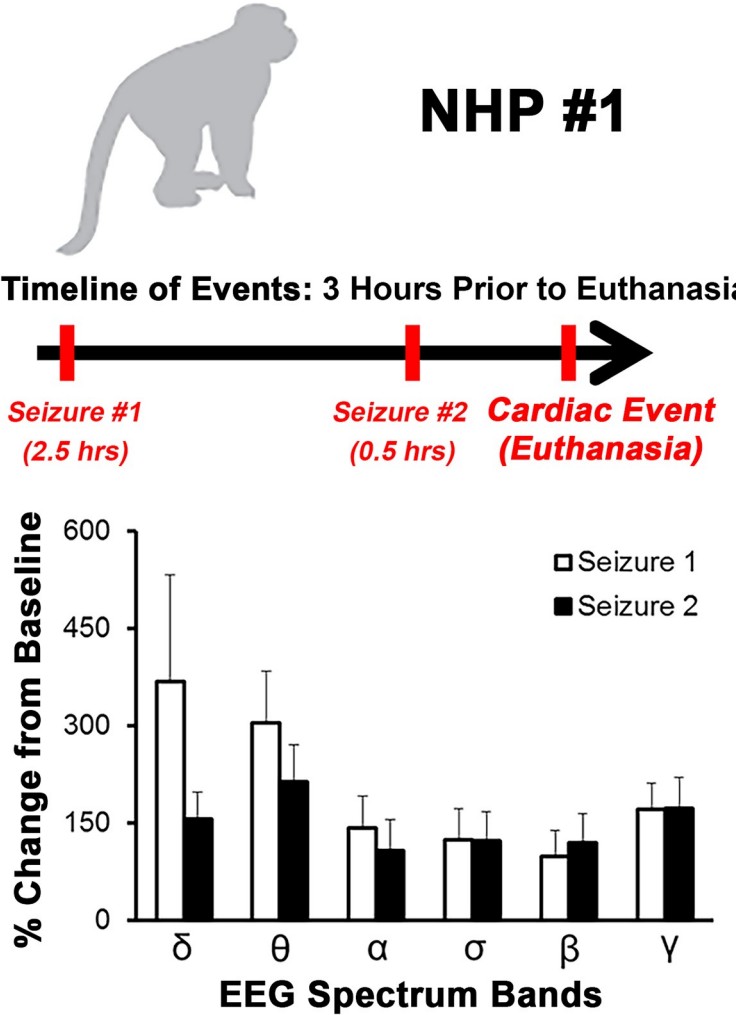

**Fig 13. Timeline of events for NHP #1 3 hrs prior to euthanasia and profile of seizure events.** Percent change in EEG waves with +/- standard deviations (error bars) are shown.

### Clinical parameters of nhp #1 three hours prior to a cardiac event

NHP#1 had experienced considerable alterations of many important physiological parameters (temperature, respiration, heart rate, blood pressure, ECG, and EEG) ~24–80 hrs prior to euthanasia. In addition, there was substantial decline in sleep and food/fluid consumption ~40–52 hrs prior to euthanasia (S1 Table). During the last 3-hrs prior to euthanasia, NHP#1 experienced two seizures ~2 hrs apart (Fig 13). Both EEG seizure profiles showed involvement of all brain waves with increases ranging from ~99% to ~368% (Fig 13). The time frame of events occurred at the shift from daytime to nighttime, when the baseline values for each parameter would naturally decline in healthy animals (Fig 14). Following the onset of the first seizure, many of the parameters remained elevated and/or increased relative to baseline. The comparison of peak respiration and heart rates to baseline ranged from 31 vs. 18 bpm and 187 vs. 65 bpm, respectively (Fig 14A and 14B). Similar peak increases were also observed for systolic and diastolic blood pressure relative to baseline with 180 vs.103 mmHg and 121 vs. 67 mmHg, respectively (Fig 14C and 14D). These elevations in parameters represent percent

# NHP #1

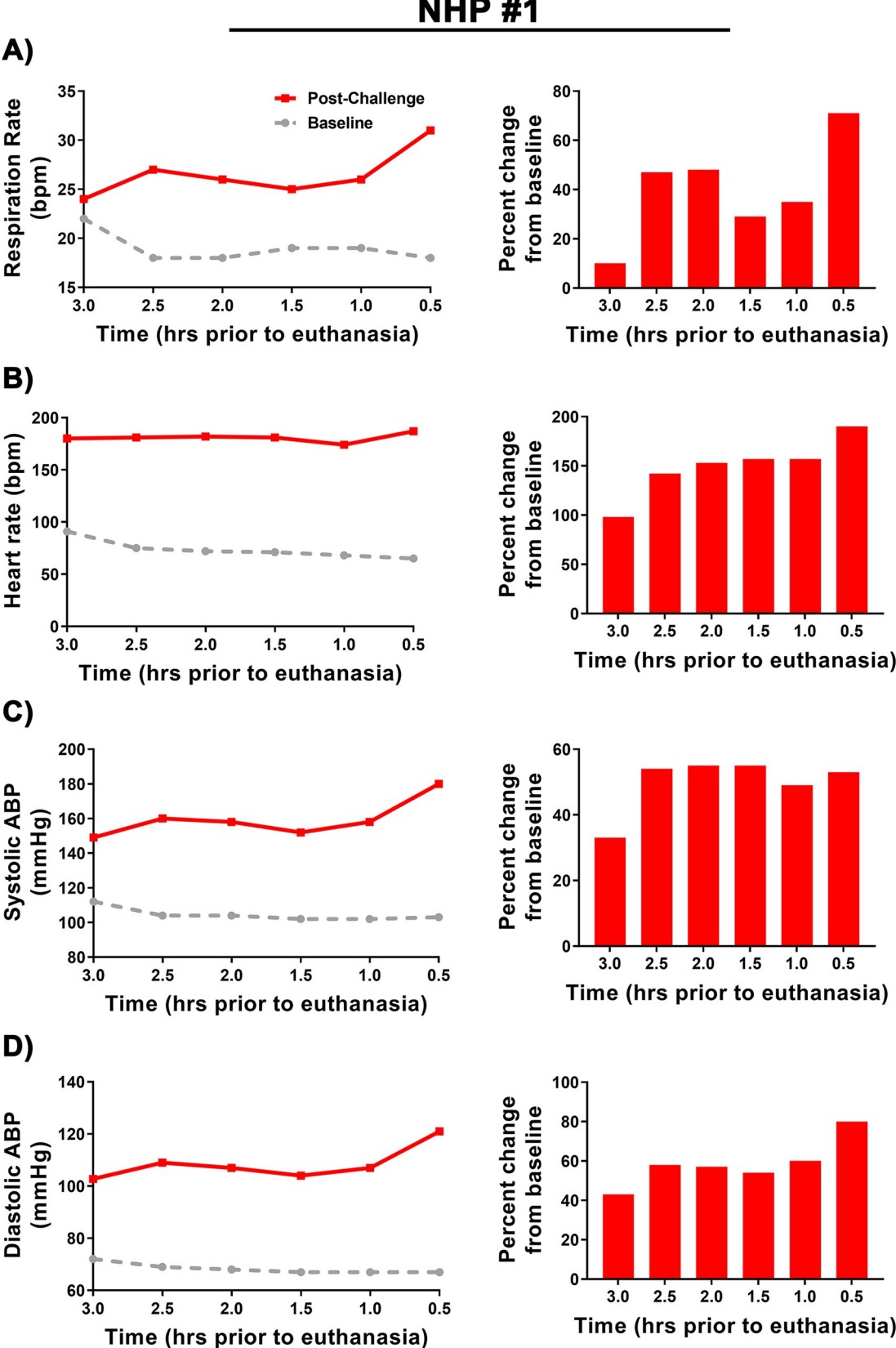

**Fig 14. Alteration in respiration, heart rate, and systolic and diastolic blood pressure of NHP #1 3 hrs prior to euthanasia.**
0.5-hr interval analysis (left) and percent change from baseline (right). The average for each time-matched 0.5 hr interval in shown in grey.

increase from baseline of 71% (respiration rate), 190% (heart rate), 55% (systolic blood pressure), and 80% (diastolic blood pressure) (Fig 14). Following the onset of two seizures, NHP #1 experienced a cardiac event characterized by non-sustained ventricular tachycardia, followed by sustained ventricular tachycardia, and finally ventricular fibrillation (Fig 15). Immediately following the cardiac event, the NHP was euthanized.

## Discussion

The susceptibility of cynomolgus macaques to North American lineage of EEEV via the aerosol route has been explored previously with two isolates, FL91-4679 and FL-93939. Both isolates were obtained from mosquito pools; *Aedes albopictus* (FL91-4679) and *Culiseta melanura* (FL-93939) [14,15]. Aerosol exposure to either isolate at $10^7$ PFU produced uniform disease and NHPs met the euthanasia criteria within ~96–130 hpi [14,15]. In this study, we utilized an isolate from the brain tissue of a fatal human case in Massachusetts in 2005 [16,18]. The data from our study are in agreement with two previous studies with all four NHPs meeting euthanasia criteria by ~106–140 hpi [14,15]. Taken together, all three studies demonstrate that low passage isolates of EEEV-NA lineage, irrespective of the isolation from mosquito and human hosts, are uniformly lethal at $10^7$ PFU dose via the aerosol route.

Two decades ago, Pratt and colleagues successfully utilized telemetry in biocontainment to demonstrate continuous monitoring of temperature in NHPs following VEEV challenge [19]. This success led to incorporation of temperature in NHP studies for category A and B pathogens including Ebola, Marburg, VEEV, EEEV, and WEEV [14,15,19–29]. However, the technology of telemetry implants has advanced considerably and is now able to monitor many important physiological parameters continuously and simultaneously in biocontainment. The current devices can measure physiological parameters at 1–1,000 Hz and produce enormous data sets ranging from 1,800 to 1,800,000 data points every 30 mins to describe a given parameter. Accordingly, the technology has substantial potential to improve animal model development particularly for Risk Group 3 and 4 agents. This is the first of its kind study in biocontainment to investigate clinical disease course of EEEV by implanting multiple devices in a single NHP to continuously and simultaneously monitor temperature, activity, respiration, heart rate, blood pressure, ECG, and EEG. All physiological parameters were altered considerably post-challenge and all four NHPs met the euthanasia criteria rapidly. However, the onset and sustained duration of each parameter differed considerably. Surprisingly, EEG was the earliest parameter to change within ~12–36 hpi in all four NHPs, followed by temperature, blood pressure, and activity/clinical signs at ~48–72 hpi, and all others ≥72 hpi.

In previous studies, only temperature and heart rate parameters have been explored [14,15]. However, the heart rate data was limited as minimal baseline day/night time data was provided and partial post-challenge data was reported [14,15]. In both previous studies, the onset of sustained fever occurred ~48 hpi with peak increase in temperature ranging from 1.8–3.5˚C, followed by a rapid decline prior to euthanasia [14,15]. The onset of sustained elevated heart rate was between ~40–72 hpi with peak heart rate increase of ~40–130 bpm [14,15]. Our temperature and heart rate data are in agreement with both of the previous studies.

Cynomolgus macaques are a prey species and to avoid the potential confounding effects of prey response elicited by cage-side human observations, we utilized 24-hr continuous remote monitoring to gain greater insight into alteration of macaque behavior and clinical

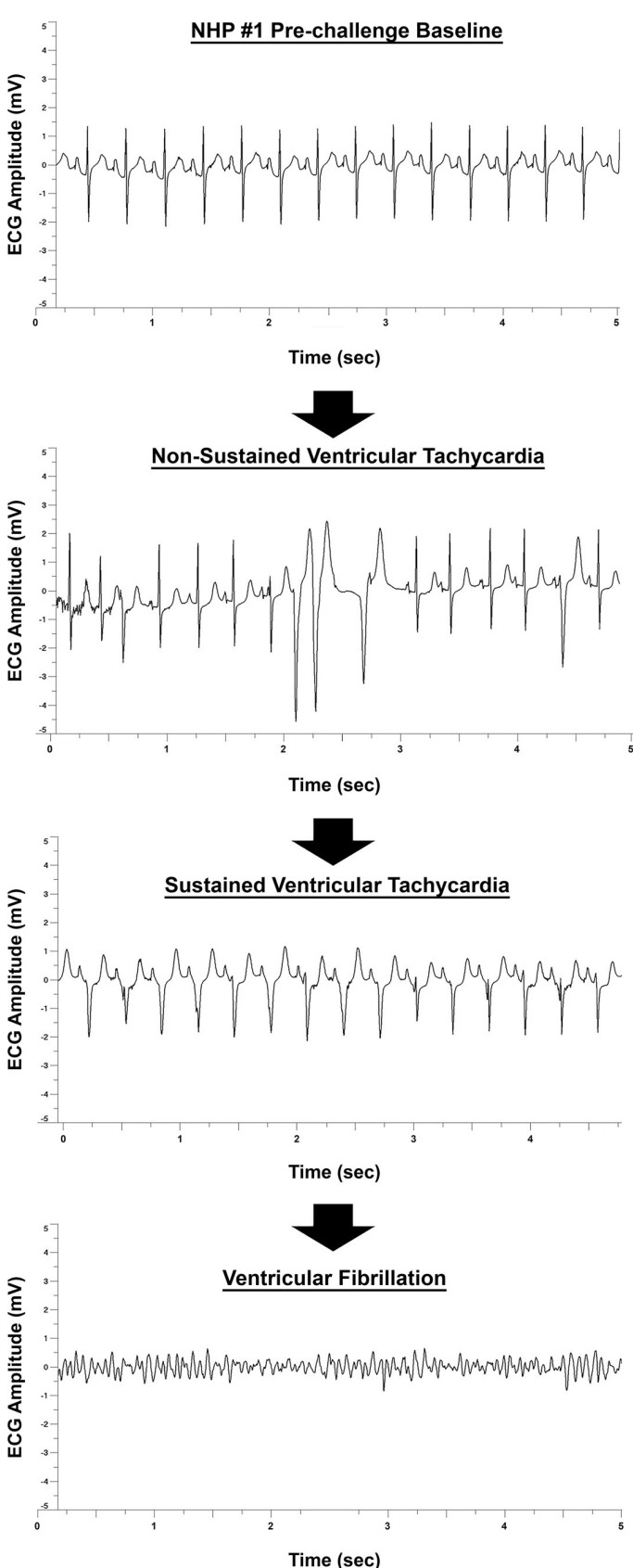

**Fig 15. ECG profile of the cardiac event in NHP #1.** Representative baseline and cardiac event ECG graphs ~5 seconds in duration are shown.

manifestations post-EEEV infection. Disturbances in the circadian rhythm post-challenge were observed as early as 24 hpi. The NHPs were observed staying awake and alert at night time with a concomitant decrease in daytime activity accompanied by short periods of sleep. The alteration in sleep/wake cycle was also accompanied by decrease in fluid and food intake, followed by a rapid progression to minimal sleep, activity, and food/fluid consumption. The substantial alteration of these parameters is likely to considerably exacerbate the observed clinical signs in the terminal phase of infection. The weakness and lack of coordination observed in the last 24 hrs prior to euthanasia may be in part due to the lack of nutrients, electrolyte imbalances, and lack of sleep. Lastly, the continuous remote observation of the NHPs enabled the study to accurately measure the onset of neurological disease. Seizures were observed in three of the four NHPs, however, the on-set of seizures was very late in the disease, ~1–3 hrs prior to euthanasia. Taken together, these data demonstrate that remote monitoring of animals can substantially enhance the understanding of natural disease progression and should be incorporated in the objective assessment of clinical disease in future NHP studies.

This study investigated ECG by measuring QTc Bazett, PR, RR, and QRS duration. These intervals are dependent on the heart rate. As the heart rate increases or decreases the intervals get shorter or longer, respectively [30]. Alterations in heart rate were observed as early as ~24 hpi and sustained increases were maintained through ~79–140 hpi. Consequently, the decrease in the intervals are consistent with an increase in heart rate. A recent study measured QT and RR intervals following EEEV infection via aerosol route [31]. A decrease in both intervals was observed following onset of severe disease [31]. Our data are in agreement with this study.

However, there were ECG abnormalities detected within the last 24 hrs prior to euthanasia. The NHPs displayed ECG abnormalities consisting of alterations in QRS and PR duration, QTc Bazett, T wave morphology, amplitude of the QRS complex, and sinoatrial arrest. These abnormalities are indicative of electrical conductivity issues in the heart and are associated with ventricular arrhythmias [32]. In addition to these abnormalities, NHP #1 experienced a critical cardiac event leading immediate euthanasia. This is the first evidence of life-threatening critical cardiac events as a consequence of EEEV infection. There are several potential explanations for the cardiac abnormalities. First, the abnormalities may be due to EEEV infection of the myocardium and/or the pericardium [33–39]. EEEV infection of the myocardium and subsequent degeneration of the spontaneously contracting cardiac muscle tissue has been reported in equines, swine, and humans [40–42]. Second, the abnormalities could be due to host inflammatory responses resulting in myocarditis and/or pericarditis [33–39]. QRS and QT prolongation, ventricular arrhythmias, and T wave morphology changes have been reported for viral infections such as Coxsackievirus, HIV, influenza A, HSV, and adenovirus [32–39]. Third, the EEEV may target important autonomic control center such as the hypothalamus, thalamus, and medulla oblongata and thus interfere with electrical conductivity of the heart. Fourth, the electrolyte imbalance due to lack of food and fluid intake prior to meeting the euthanasia criteria may exacerbate any of the explanations outlined above.

Many human EEEV infection cases are often misdiagnosed and/or diagnosed at the onset of severe symptoms, and accordingly a detailed clinical disease course is not available for human infection [40,43–57]. One important goal of animal model development is to gain insights into progression of brain abnormalities leading to encephalitis. To achieve this goal, we investigated EEG as a potential tool enabling continuous monitoring of the brain electrical activity following challenge. EEG has been utilized previously to monitor human EEEV,

WEEV, and VEEV infections [43–45,47,49–51,53,54,56,58–61]. The limited human EEG data shows diffuse slowing particularly of delta, theta, and alpha and gamma waves [43–45,47,49–51,53,54,56,58–61]. Similar to human data, severe diffuse slowing was observed in all four NHPs post-EEEV challenge. The data in the present study is in agreement with previous human EEG data. However, in addition to diffuse slowing, other gross abnormalities were also observed. First, brain waves of all four NHPs displayed rapid and extreme fluctuations. In NHP #3, a profound decrease in all brain frequencies of up to nearly 99% was observed at ~94–108 hpi, followed by an increase of over 600, 650, 2,500 and 6,300% in alpha, sigma, beta, and gamma frequencies, respectively, for a duration of ~4–12 hrs. Similarly, NHP #4 displayed an increase of ~230 to 3,500% in delta, theta, alpha, and/or sigma frequencies at ~84–108 hpi followed by a near complete decline in all four waves for a duration of ~30 hr. Second, there was a profound decrease in gamma frequencies of nearly 99% during the daytime that continued for up to ~10–15 hrs. Third, the presence of brain waves associated with sleep (delta, theta, and alpha waves) in awake NHPs was observed. All three brain abnormalities are a sign of significant brain injury and have been observed in human cases of viral encephalitis such as Japanese encephalitis virus and Herpes simplex virus [62–67]. Further studies are underway to characterize these profound alterations in the brain waves as well as the underlying mechanism/s responsible for these abnormalities.

Traumatic brain injury (TBI) is defined as a non-degenerative and non-congenital insult to the brain that results in temporary or permanent impairment of cognitive and physical functions [68]. All four NHPs exhibited many signs associated with TBI such as disturbances in circadian rhythm, food/fluid consumption, inability to initiate or maintain a normal sleep pattern, decrease overall activity, increased slow wave (delta, theta, and alpha) activity while awake, and neurological signs [68–72]. These data strongly suggest that EEEV infection via aerosol route can rapidly (~12–70 hpi) induce many signs of severe TBI. The potential mechanism/s that underlie such rapid induction of TBI signs require further exploration.

The potential route of virus dissemination following aerosol infection has not been explored previously [14,15]. The infectious virus in the present study was either minimal or not detected in the periphery. In contrast, high level of infectious virus (>6.0 $\log_{10}$ PFU/g) was detected in the olfactory bulb and the central nervous system (CNS) in all animals at the time of euthanasia. In addition, the physiological parameters measured in this study such as heart and respiration rates, blood pressure, temperature, sleep/wake cycle, and hunger/satiety are controlled by the autonomic nervous system (ANS). The rapid (~24–50 hpi) and considerable alteration of these functions suggest that EEEV infection via the aerosol route likely enables direct access to the ANS via the neuronal projections between the olfactory bulb and the critical control centers such as the thalamus, hypothalamus, and the brainstem. Taken together these data suggest that following aerosol infection the virus spreads and infects the CNS and ANS producing injury to substantially disrupt animal behavior and the control of critical physiological parameters to produce severe disease. However, this hypothesis requires further investigation to elucidate the potential mechanism.

The clinical scores in of NHPs displayed two distinct patterns, a rapid rise (NHPs #1–2) or biphasic pattern with an initial rise followed by a decline (NHPs #3 and 4). The clinical score is comprised of neurological disease, temperature, and responsiveness. The initial rise in clinical scores of NHPs #3 and 4 were mainly due to alteration in both temperature and responsiveness. However, the temperature declined between 90–110 hpi and the responsiveness improved to yield lower clinical scores. Nonetheless, both NHPs rapidly progressed to meet the euthanasia criteria (clinical score = 10). Surprisingly, the temperature rapidly declined to below baseline and both NHPs exhibited hypothermia ~1–3 hrs prior to euthanasia. These data suggest both NHPs likely were unable to regulate body temperature and support the

hypothesis of ANS dysregulation above. The data also highlight the rapid progression to terminal disease of EEEV infected NHPs.

Neutralizing antibodies are generally considered to be a correlate of protection against alphavirus infection [73–78]. However, two NHPs in this study had neutralizing antibody at the time of euthanasia. One potential explanation is that EEEV infection and subsequent injury to the brain tissue is rapid and profound via the aerosol route, and by the time the neutralizing antibody response is generated it has minimal or no efficacy. This is supported by previous vaccine studies in cynomolgus macaques with similar rapid aerosol infection kinetics of EEEV [13,15]. In both studies, NHPs generated neutralizing antibody responses post-vaccination and yet met the euthanasia criteria following challenge [13,15].

Previous studies in cynomolgus macaques with lethal aerosol EEEV have focused mainly on five parameters (temperature, virus quantitation, hematological parameters, clinical disease, and lethality) for NHP model development [13–15]. This study provides six additional parameters for countermeasure evaluation including activity, respiration and heart rates, blood pressure, ECG, and EEG in a lethal challenge model. For this first study, we utilized the most lethal of the three encephalitic alphaviruses, EEEV. Whether a similar alteration of these parameters following aerosol challenge in NHPs can be produced with WEEV and VEEV or a sub-lethal EEEV dose requires further investigation.

The FDA approval of medical countermeasures for Risk Group 3 and 4 agents will rely on the U. S. Food and Drug Administration's Animal Rule (21 CFR 601.90), which allows the utilization of animals in pivotal efficacy studies to support licensure in lieu of human efficacy studies. The measurement of clinical parameters via telemetry offers several advantages for animal model development in this regard. First, the technology enables measurements of important clinical parameters that are relevant to humans. Second, it enables identification of multiple parameters for rapid animal model development and countermeasure evaluation in event of a natural outbreak and/or bioterror event. The additional parameters may be particularly appealing for investigating and/or refining animal model development for partially lethal or non-lethal agents such as WEEV and MERS. Furthermore, the high sampling rate of the devices enables accurate and real-time (hourly/daily) evaluation of countermeasures. Third, it can identify potential side effects of a countermeasure prior to its utilization in human clinical trials. These are substantial advantages of utilization of advanced telemetry in NHP animal models for Risk Group 3 and 4 agents and require further investigation.

In summary, we utilized state of the art telemetry to investigate important physiological parameters in the cynomolgus macaque model following EEEV aerosol challenge. All parameters measured, including 6 never-before examined including brain waves, were substantially and rapidly (~12 hpi) altered post-challenge. These alterations were the earliest documented signs of disease that were not readily observable without the use of physiological radio-telemetry devices, and add possible additional endpoints to future efficacy experiments for EEEV in an NHP model. This is the first detailed disease course of EEEV in an NHP model and the parameters identified will improve animal model development and countermeasure evaluation.

## Disclosure statement

The views expressed in this article are those of the authors and do not reflect the official policy or position of the U.S. Department of Defense, or the Department of the Army.

## Supporting information

**S1 Fig. Clinical scores of NHPs post-EEEV challenge.** Following aerosol challenge all NHPs were monitored daily and NHPs with a total score ≥10 met the euthanasia criteria.
(TIF)

**S2 Fig. QRS duration and QTc Bazett in NHPs pre- and post-EEEV challenge.** Pre-challenge baseline QRS duration and QTc Bazett are shown in grey (A). All NHPs were continuously monitored pre- and post-challenge. Pre-challenge baseline QRS duration and QTc Bazett were measured for five day/night cycles and a 0.5-hr interval baseline average was calculated by averaging raw data of five time-matched day or night time intervals. Forty-eight 0.5-hr interval averages were used to construct baselines for QRS duration and QTc Bazett for a day/night cycle and are shown as a grey line (A). Post-challenge values within $\leq$3 standard deviations (SD) are indicated with (♦), >3 SD above baseline are indicated with (♦), and >3 SD below baseline are indicated with (♦).
(TIF)

**S3 Fig. RR and PR duration in NHPs pre- and post-EEEV challenge.** Pre-challenge baseline RR and PR duration are shown in grey (A). All NHPs were continuously monitored pre- and post-challenge. Pre-challenge baseline RR and PR duration were measured for five day/night cycles and a 0.5-hr interval baseline average was calculated by averaging raw data of five time-matched day or night time intervals. Forty-eight 0.5-hr interval averages were used to construct baselines for RR and PR duration for a day/night cycle and are shown as a grey line (A). Post-challenge values within $\leq$3 standard deviations (SD) are indicated with (♦), >3 SD above baseline are indicated with (♦), and >3 SD below baseline are indicated with (♦).
(TIF)

**S1 Table. Alteration in NHP behavior and onset of neurological signs post-EEEV challenge.** NHP behavior comprised of food/fluid intake, sleep, activity, and onset of seizures were monitored pre- and post-EEEV challenge. $\downarrow$ = modest decline, $\downarrow\downarrow$ = moderate decline, and $\downarrow\downarrow\downarrow$ = severe decline. $\uparrow$ = modest increase.
(TIF)

**S2 Table. Neutralizing antibody response in terminal samples.** Neutralizing antibody was measured via $PRNT_{80}$ assay. The limit of detection in $PRNT_{80}$ assay is indicated by italic font (*<1:20*). All samples were analyzed three times in the $PRNT_{80}$ assay.
(TIF)

**S3 Table. Summary of fever data in NHPs.** Fever hours is calculated as the sum of the significant temperature elevations. $\Delta T_{max}$ = maximum change in temperature.
(TIF)

## Author Contributions

**Conceptualization:** William D. Pratt, Margaret L. Pitt, Farooq Nasar.

**Formal analysis:** Franco D. Rossi, Michael V. Accardi, Simon Authier, William D. Pratt.

**Funding acquisition:** Margaret L. Pitt, Farooq Nasar.

**Investigation:** John C. Trefry, Franco D. Rossi, Michael V. Accardi, Brandi L. Dorsey, Thomas R. Sprague, Suzanne E. Wollen-Roberts, Joshua D. Shamblin, Adrienne E. Kimmel, Lynn J. Miller, Anthony P. Cardile, Darci R. Smith, Sina Bavari, Simon Authier, William D. Pratt, Farooq Nasar.

**Methodology:** John C. Trefry, Franco D. Rossi, Michael V. Accardi, Simon Authier, William D. Pratt, Farooq Nasar.

**Project administration:** John C. Trefry, Margaret L. Pitt, Farooq Nasar.

**Resources:** Pamela J. Glass, Crystal W. Burke.

**Supervision:** Margaret L. Pitt, Farooq Nasar.

**Writing – original draft:** John C. Trefry, Farooq Nasar.

**Writing – review & editing:** John C. Trefry, Franco D. Rossi, Michael V. Accardi, Brandi L. Dorsey, Thomas R. Sprague, Suzanne E. Wollen-Roberts, Joshua D. Shamblin, Adrienne E. Kimmel, Pamela J. Glass, Lynn J. Miller, Crystal W. Burke, Anthony P. Cardile, Darci R. Smith, Sina Bavari, Simon Authier, William D. Pratt, Margaret L. Pitt, Farooq Nasar.

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
