## [Decision Letter · Decision Letter 0]

29 Dec 2020

Dear Dr Nasar,

Thank you very much for submitting your manuscript "The Utilization of Advance Telemetry to Investigate Important Physiological Parameters Including Electroencephalography in Cynomolgus Macaques Following Aerosol Challenge with Eastern Equine Encephalitis Virus" for consideration at PLOS Neglected Tropical Diseases. As with all papers reviewed by the journal, your manuscript was reviewed by members of the editorial board and by several independent reviewers. In light of the reviews (below this email), we would like to invite the resubmission of a significantly-revised version that takes into account the reviewers' comments. 

We cannot make any decision about publication until we have seen the revised manuscript and your response to the reviewers' comments. Your revised manuscript is also likely to be sent to reviewers for further evaluation.

Sincerely,

Alain Kohl

Associate Editor

Ann Powers

Deputy Editor

Reviewer's Responses to Questions

**Key Review Criteria Required for Acceptance?**

**Methods**

-Are the objectives of the study clearly articulated with a clear testable hypothesis stated?

-Is the study design appropriate to address the stated objectives?

-Is the population clearly described and appropriate for the hypothesis being tested?

-Is the sample size sufficient to ensure adequate power to address the hypothesis being tested?

-Were correct statistical analysis used to support conclusions?

-Are there concerns about ethical or regulatory requirements being met?

Reviewer #1: The methods are clearly described and the study design is adequate with the small numbers of animals (n=4) for the highly detailed and focussed set of analyses described. Virologists reading this article will appreciate the strength and depth of the physiological measurements and the extended knowledge in this area. However, these detailed changes described are taken or assumed to be due to a consistent virus inoculum being delivered and hence the changes observed are consistent with a varied host response to virus. 7.0log10 Pfu by aerosol challenge was administered and data provided suggested dose received by the animals. Is there any sequence or virological data relating to the viral stock used? This would be useful to include even if re-iterative. Aerosol challenges often result in several log reductions in effective viral titre administered – can the authors comment on this and is there a dose effect in delivering different amounts of viral challenge with this system?

Reviewer #2: The study design of this manuscript is fairly robust, especially given the inherent limitations of NHP models. The overall aim was to assess technological improvements in telemetry to investigate key aspects of EEEV parthenogenesis (temperature, activity, respiration heart rate, blood pressure, ECG and EEG). The low sample size and the use of the same animals as baseline controls is not ideal, but I'm happy that sufficient number of animals were used to draw statistically meaningful conclusions. One issue i have is the lack of justification for the use of NHPs to test the telemetry technology and this should be included in the text i.e why are NHP required? could mice/rats have been used instead?. As a final thought on experimental design, I wasn't entirely sure why the authors infected the NHPs via the aerosol route, given that the primary route of infection is through the skin during a mosquito blood meal. I may have missed it, but an additional sentence to justify this would improve the rationale of the manuscript.

Reviewer #3: The objective of the study is clear and nicely presented.

All ethical and regulatory requirements were met.

In this article, John Trefy et all from the teams of Farooq Nasar at the USAMRIID center demonstrated the usefulness of telemetry to live follow-up of NHP exposed to an encephalitic inducing virus. 

They thus provide more precise and useful clinical parameters associated to the course of this severe and acute disease. Specifically, they shown that beside the expected direct effect of EEEV on brain function, a very rapid and parallel alteration of cardiac function might be a direct cause of death in EEEV exposed patient that thus might provide a new clue on this deadly disease.

Specifically, the authors provide a new system able to measure in real time both temperature, activity, respiration, heart rate, blood 132 pressure, electrocardiogram (ECG), and electroencephalography (EEG). 

Altogether the presentation is of very high quality, easy to read and results are significant despite the relative low number of animals studied (4). The description is clear and meaningful and the routine biological methods associated to assess the disease advancement and viral replication and dissemination within the infected animals are fully described. 

The author chose to use a relatively high viral load to infect the animals and thus all the animals were killed by the disease (or more exactly reach the humane endpoint that induced euthanasia). One of the most interesting point is the direct involvement of cardiac failure as immediate cause of severity.

The objective of this work to demonstrate the usefulness of advanced and invasive telemetry to provide new clues about a long-time studied model of Cynomolgus Macaques exposed to an Aerosol Challenge with Eastern Equine Encephalitis Virus is thus reached. 

However, the main new method presented here (i.e. the telemetry system) is poorly described and only brand mark and analytic performance were provided. Personally, I wonder if the very high cost of such a system deserved to be provided. Even if the specific method of aerosol challenge is already well described, the time cost and method used to implement the animals with the telemetry device should be provided here.

As an example, how long time is needed in order the animals recovered from the device implantation, is there a difference or not between animals of 3kg or 9 kg as indicated. I imagine the female and male differences as the sexual dysmorphism is high in this macaque species. Could you please provide this information in the table A) within figure 2.

At least, references to the device provider reports or vet article describing these procedures might be of interest for the non-specialized reader to highlight the work needed here. Could the device be re-used ? are all animals infected at the same time or successively ?

**Results**

-Does the analysis presented match the analysis plan?

-Are the results clearly and completely presented?

-Are the figures (Tables, Images) of sufficient quality for clarity?

Reviewer #1: For small numbers of animals, it is instructive to relate the limited but valuable virological data generated (supplementary figure 2) directly back to the clinical observations. Were earlier serum or plasma samples taken that would confirm on-set or presence of a viraemia in this model. Were there any cytokine changes noted? Some reference to these virological and immunological variables would be instructive. 

It would be extremely valuable to relate the infectious virus titre data generated more specifically to the clinical observations. For example, NHP#1 is the only animal to have a detectable viraemia in serum and plasma (approx. 3.5 log 10Pfu). The other three are undetectable – this on the face of it would seem a significant observation, at least worthy of a brief comment. NHP#3 has 4 logs of virus in the heart but the other three are undetectable – how does this relate to heart output measures? - again no direct comment by the authors. Linking these observations would hugely enrich the manuscript. 

Infectious virus detected in the olfactory bulb in all 4 and in the brain is at least worthy of a comment on the route of virus trafficking to the brain via an aerosol administration – is there any pathology data that would inform these events? Perhaps not the focus of this study and beyond current scope but would be very interesting follow-up study if the materials existed or were collected. Are there any PCR data for this model that could be generated/included, - virus-specific cell-associated RNA quantification would be a useful additional biomarker to relate across to the infectious virus data. 

Hence, it would seem highly relevant to relate these limited, but nonetheless informative virological findings back/across to the physiological data. To enable this I would recommend moving Supp Fig 2 into the main body of the manuscript and integrating these data into the descriptions of the results and work into the discussion as appropriate.

Reviewer #2: Results and analysis were generally clear however there are a few outstanding issues:

Section "EEEV challenge study" - It would improve the entire results section if a brief description of clinical signs was added here to help the reader interpret the results. Supp. figure 1 is very confusing, the sampling is poorly explained, i.e. why were the animals sampled different number of times depending on the dpi? I think it would be more appropriate to display that as hours post infection, in line with the other figures in the manuscript. An attempt should also be made to discuss the bi-phasic disease profile in NHP 3 and 4, is it expected that NHP 4 would have recovered to have 0 clinical score before euthanasia? or is this a limitation of the scoring system itself? 

I have no issues with the "alteration in animal behaviour" section, I think one of the strengths of this manuscript is the ability to limit external interference. 

 "Detection of Infectious EEEV in Various Tissues" - Although interesting, due to the low N number and spread of results (also the difference in endpoints), the biological relevance of the tissue tropism is difficult to interpret. I would like to see the relevance and limitations of the viral loads and tissue tropism discussed in the discussion section, alternatively, as this is supplementary, perhaps consider making this section more succinct and amalgamating it with "EEEV Challenge Study" as tissue tropism isn't a strong focus of this manuscript. 

"Neutralizing Antibody Response" - Figure 2C would be more appropriate as a supplemental figure, It is important prior to infection to demonstrate the NHPs did not have neutralising Ab, however more work is required to characterise the adaptive immune response, especially as the time post infection is too early for class switching, high affinity neutralizing antibodies. Additionally, the number of repeats for the PRNT assay should be stated in the legend clearly, to ensure this isn't an artefact. 

"Temperature." - This is where this manuscript really starts to assess the telemetry technology, where i believe it's strengths lie. I have a few questions to be clarified: The baseline for 3A (this is relevant for figure 4-8) is confusing, is the baseline taken for the whole 144 hours pre infection? or is it repeating measurements over a 24 hours? because the grey line looks remarkably consistent for a parameter such as temperature. to Avoid confusion and increase transparency, figure 3A should be presented like 3B i.e. with the X-axis starting at -96hpi and finishing at 144hpi

"respiration" - same as above, interesting data which would be improved by tweaking the presentation of the results of A to mirror that of B. 

"Activity - blood pressure" - Same as above, consider changing presentation

"ECG" -Caveat being, this isn't my area of expertise, but it looks well presented to me.

Reviewer #3: Result are clear and meaningfull

**Conclusions**

-Are the conclusions supported by the data presented?

-Are the limitations of analysis clearly described?

-Do the authors discuss how these data can be helpful to advance our understanding of the topic under study?

-Is public health relevance addressed?

Reviewer #1: With any virological NHP challenge study, however, there will be inevitable variations in the replication dynamics and host response to the virus. The authors could make a general point as to why Chinese cynomolgus macaques are the model of choice here and perhaps compare to other cynomolgus macaque species (eg Mauritian or Indonesian), as to why select one species over another. Would different outcomes be anticipated in such circumstances? Integrating combined physiological outcomes with the virological data presented in the discussion would improve the paper. 

Public health relevance is well stated and teh authors describe how this model relates to this.

Reviewer #2: The conclusions of this manuscript are clearly stated and nicely put into context of the work that has previously been undertaken. Moreover, a satisfactory job has been made to discuss the findings in the context of how telemetry could be useful for future studies. One minor issue is that portions of interesting data are presented as if this is a preview for a later, more in-depth study, i Would like to see statements such as "The underlying cause/s of ECG abnormalities are currently under investigation", and "sub-lethal EEEV dose remains to be seen and will be investigated in future

studies" removed or reworded.

Reviewer #3: Conclusion are supported by the data.

Limitation per se are not described and may be more due to the money cost than from the scientific work relevance that is of high standard quality.

Public health relevance was carefully addressed.

In addition, could the author may indicate how they investigate the underlying mechanisms as stated line 458-459 and 486-487 (heart and brain structure evaluation and/or virus research). Will this only be done with the materials from the described 4 animals or this will involve historical samples (and which type of samples) from previous study of the very same model? Even, if out of the main scope of this article, how the author will take in account the variability of the death timing (here between 4 and 6 days)?

**Editorial and Data Presentation Modifications?**

Reviewer #1: Move supplementary figure 2 into the bulk of the manuscript and integrate these observations into the descriptions of the physiological data/measures.

Reviewer #2: Figure 1 - indicate on figure where data is collected i.e. if continuous, state that in the figure. 

Figure 2 - state number of technical repeats in PRNT assay in the figure legend

Figure 3 through 8 - be clear how the baseline was calculated in A, consider changing the presentation of "A" to match "B"

Figure 9 - no comment

Figure 10 through 11 - not my area of expertise but it is easy to interpret

Figure 13 - again, confusing how and when the baseline is calculated? are they matched to the same time on different days? clarify in the figure legend 

Supp.1 - discuss limitations of clinical score in the text, is it expected that they would recover immediately prior to euthanasia? 

Supp.2 - consider removing CSF in this figure, the incomplete data set is confusing 

Supp. 3 through 4 - consider changing style to make the baseline clearer

Reviewer #3: Minor remarks:

1) The term Chinese origin for cynomolgus is more a producer brand as the natural range of Macaca fascicularis do not reach China mainland. Thus, an indication of the genetic background of these animals might be of interest as at least 3 main groups are well known: from 1) Vietnam and Asia continental area to 2) Indonesia archipelago and finally 3) in Philippian archipelago. And this, when we take not in account the very specific case of the Mauritius sub-group that is largely used in biomedical research.

2) please provide a reference for the aerosol exposition method and viral titer evaluation for the exposition in the result lines.

3) Could you explain (or have an hypothesis) the major modification for the NHP#1 at -16 pre-challenge in figure 10 ? There were also some blue in the baseline for NHP#4

**Summary and General Comments**

Reviewer #1: Trefry and colleagues describe the use of advanced telemetry techniques to monitor a range of physiological read-outs in a Chinese cynomolgus macaque model of Eastern equine encephalitis virus following aerosol administration. This a very comprehensive description of the range of physiological measurements that can be undertaken the specialist facility at USAMRIID. The large number of Figures summarising and describing these data reflect the range of measures that have been undertaken. Development and refinement of models to investigate the possible outcomes of virus infection in understanding the pathogenesis of the virus and to develop suitable models to evaluate intervention strategies are hugely valuable. 

As outlined in Results the infectious virus load data needs to brought into the paper more.

Better integration of these virological observations to the clinical/physiological measures taken would enhance this paper enormously. Do the authors intend to follow up on brain pathology? This would seem an obvious next step and to maximise the resources of this hugely valuable study which is beyond the scope of most institutions.

Reviewer #2: There is a few outstanding issues with the presentation of this manuscript which can be easily improved with extra clarification in the text, including more robust justification for using NHPs, changes to how baselines are presented and some amalgamation of less relevant data sets i.e. antibody response and tissue tropism. Generally, a good attempt has been made to put this data into the broader context of the literature, however, much of the manuscript feels incremental, and i would have loved to see the inclusion of work relating to the infection of the autonomic nervous system, the cardiac event of NHP1 and the mechanisms of traumatic brain injury.

Reviewer #3: no more comments

PLOS authors have the option to publish the peer review history of their article (what does this mean?). If published, this will include your full peer review and any attached files.

Reviewer #1: No

Reviewer #2: No

Reviewer #3: No
---

## [Decision Letter · Decision Letter 1]

29 Apr 2021

Dear Dr Nasar,

We are pleased to inform you that your manuscript 'The Utilization of Advance Telemetry to Investigate Important Physiological Parameters Including Electroencephalography in Cynomolgus Macaques Following Aerosol Challenge with Eastern Equine Encephalitis Virus' has been provisionally accepted for publication in PLOS Neglected Tropical Diseases.

Best regards,

Alain Kohl

Associate Editor

Ann Powers

Deputy Editor

Reviewer's Responses to Questions

**Key Review Criteria Required for Acceptance?**

**Methods**

-Are the objectives of the study clearly articulated with a clear testable hypothesis stated?

-Is the study design appropriate to address the stated objectives?

-Is the population clearly described and appropriate for the hypothesis being tested?

-Is the sample size sufficient to ensure adequate power to address the hypothesis being tested?

-Were correct statistical analysis used to support conclusions?

-Are there concerns about ethical or regulatory requirements being met?

Reviewer #1: (No Response)

Reviewer #2: All my original issues with the methods have been dealt with by the authors sufficiently.

Reviewer #3: As this article is a proof of concept of the efficiency of advance telemetry to provide very detailled analysis of physiological parameters, the described results in 4 animals is sufficient to answer perfectly to the objective and thus did not need any statistical support and in the current stage I have no concerns about ethical point of view.

**Results**

-Does the analysis presented match the analysis plan?

-Are the results clearly and completely presented?

-Are the figures (Tables, Images) of sufficient quality for clarity?

Reviewer #1: (No Response)

Reviewer #2: I appreciate the effort that has been put into rearranging the structure of the figures and the additional text has clarified all of my concerns/issues.

Reviewer #3: Yes

**Conclusions**

-Are the conclusions supported by the data presented?

-Are the limitations of analysis clearly described?

-Do the authors discuss how these data can be helpful to advance our understanding of the topic under study?

-Is public health relevance addressed?

Reviewer #1: (No Response)

Reviewer #2: My main issues with the interpretation of the data have been addressed. Alterations to the text, particularly the last paragraph, also emphasise the novelty and importance of the manuscript, which i felt was lacking on the first submission.

Reviewer #3: Yes all points are clarified in this revised version.

**Editorial and Data Presentation Modifications?**

Reviewer #1: (No Response)

Reviewer #2: I'm happy that the authors have made substantial changes to the the data presentation that improves the readability and interpretation of the manuscript.

Reviewer #3: -

**Summary and General Comments**

Reviewer #1: (No Response)

Reviewer #2: Sufficient changes to the text and the presentation of the data have been made to this manuscript. I'm satisfied the authors have rebutted all my issues/concerns.

Reviewer #3: - nice work

PLOS authors have the option to publish the peer review history of their article (what does this mean?). If published, this will include your full peer review and any attached files.

Reviewer #1: No

Reviewer #2: **Yes: **Steven R Bryden

Reviewer #3: No

---

## [Editor Report · Acceptance letter]

14 Jun 2021

Dear Dr Nasar,

We are delighted to inform you that your manuscript, "The Utilization of Advance Telemetry to Investigate Important Physiological Parameters Including Electroencephalography in Cynomolgus Macaques Following Aerosol Challenge with Eastern Equine Encephalitis Virus," has been formally accepted for publication in PLOS Neglected Tropical Diseases.

Best regards,

Shaden Kamhawi

co-Editor-in-Chief

Paul Brindley

co-Editor-in-Chief
